# Learning precise spatiotemporal sequences via biophysically realistic learning rules in a modular, spiking network

Ian Cone[1,2], Harel Z Shouval[1]*

[1]Neurobiology and Anatomy, University of Texas Medical School at Houston, Houston, TX, United States; [2]Applied Physics, Rice University, Houston, TX, United States

**Abstract** Multiple brain regions are able to learn and express temporal sequences, and this functionality is an essential component of learning and memory. We propose a substrate for such representations via a network model that learns and recalls discrete sequences of variable order and duration. The model consists of a network of spiking neurons placed in a modular microcolumn based architecture. Learning is performed via a biophysically realistic learning rule that depends on synaptic 'eligibility traces'. Before training, the network contains no memory of any particular sequence. After training, presentation of only the first element in that sequence is sufficient for the network to recall an entire learned representation of the sequence. An extended version of the model also demonstrates the ability to successfully learn and recall non-Markovian sequences. This model provides a possible framework for biologically plausible sequence learning and memory, in agreement with recent experimental results.

*For correspondence:
harel.shouval@uth.tmc.edu

Competing interests: The authors declare that no competing interests exist.

## Introduction

So long as time flows in one direction, nature itself is fundamentally sequential. To operate in this reality, the brain needs to think, plan, and take action in a temporally ordered fashion. When you sing a song, hit a baseball, or even utter a word, you are engaging in sequential activity. More accurately, you are engaging in sequential recall of a learned activity – your actions not only have 'a' temporal order and duration but 'the' temporal order and duration which you learned. Hence, the question of how sequence representations are learned, stored, and recalled is of fundamental importance to neuroscience. Recent evidence has shown that such learned representations can exist in cortical circuits (*Gavornik and Bear, 2014*; *Xu et al., 2012*; *Cooke et al., 2015*; *Eagleman and Dragoi, 2012*; *Yin et al., 2008*), begging the question: through what sort of circuits and learning paradigms can these representations arise?

To address these questions, we introduce a modular spiking network that can robustly learn and recall both the order and duration of elements in a sequence, via a local and biophysically realistic eligibility trace-based learning rule. Although the parts of the model's construction are based upon recent experimental observations in visual cortex (*Gavornik and Bear, 2014*; *Xu et al., 2012*; *Cooke et al., 2015*), utilizing observed cell types (*Shuler and Bear, 2006*; *Liu et al., 2015*; *Chubykin et al., 2013*) and laminar structure (*Potjans and Diesmann, 2014*; *Binzegger et al., 2009*), many of its key aspects (modularity, heterogenous representations) are illustrative of general principles of sequence learning. The ability of the network to internally learn and recall both duration and order, along with its use of a local learning rule that bypasses the need for constant and explicit targets, differs from most historical and contemporary models of sequence learning (*Fiete et al.,*

*2010*; *Pereira and Brunel, 2019*; *Klos et al., 2018*; *Jun and Jin, 2007*; *Liu and Buonomano, 2009*; *Maes et al., 2020*; *Murray and Escola, 2017*; *Martinez et al., 2019*; *Rajan et al., 2016*; *DePasquale et al., 2018*; *Laje and Buonomano, 2013*; *Sussillo and Abbott, 2009*; *Nicola and Clopath, 2017*). We also present an extended formulation of the model, which is capable of learning and recalling sequences with non-Markovian (i.e. history-dependent) transitions.

A variety of different models have been proposed to account for the representation and learning of sequences. Most of these models fall into one of two classes: chain structures or recurrent neural networks (RNNs). Chain structure models of neural sequence learning operate in a method akin to synfire chains (*Abeles, 1991*) – that is, representations of different individual stimuli are linked together, in order, via feed-forward synaptic connections (*Fiete et al., 2010*; *Pereira and Brunel, 2019*; *Klos et al., 2018*; *Jun and Jin, 2007*; *Liu and Buonomano, 2009*). These models can be formulated either via an explicit neuron to neuron chain or via an implicit embedding in a random network. While such a chain-like structure can readily encode order, there is nothing internally encoding start times, stop times, or durations of the individual elements of the sequence. Some models use the resulting activity chain as a temporal basis or 'clock', upon which elements and their durations can be learned at the level of the output, but the fragility of the chain itself can make such output representations very sensitive to noise (*Liu and Buonomano, 2009*). Other models have attempted to address these issues via ad hoc solutions such as variable adaptation time constants (*Martinez et al., 2019*), but these typically require the network to have a priori information about the sequence it will be representing.

RNNs are another common class of sequence learning models (*DePasquale et al., 2018*; *Laje and Buonomano, 2013*; *Sussillo and Abbott, 2009*; *Nicola and Clopath, 2017*). Unlike explicit feed-forward chain models, RNNs learn complex sequences by leveraging rich, dynamical representations to approximate target outputs. RNNs are fully capable of encoding duration and order, and can embed multiple sequences at once. However, common learning rules for these models, such as backpropagation through time (BPTT) or FORCE, are biologically unrealistic, as they require some combination of non-local information, precise and symmetrical feedback structures, and/or explicit feedback about the targets at every time point (*Whittington and Bogacz, 2019*). Interestingly, recent work has shown that networks can under some conditions learn inputs via random feedback connections rather than backpropagation (*Lillicrap et al., 2016*), but this random feedback is less effective than BPTT for learning sequences with long time scales (*Murray, 2019*).

Experimentally, the expression of sequences in the brain often follows a compressed encoding, in which order is represented but duration is not, and sequences are replayed with a time scale dictated by the intrinsic time scales of the circuit (*Foster and Wilson, 2006*; *Skaggs and McNaughton, 1996*; *Davidson et al., 2009*; *Ji and Wilson, 2007*). Such results can be accounted for by chain-like models. However, there are also many cases where neural sequences are learned and replayed at or near their behavioral time scale (*Gavornik and Bear, 2014*; *Louie and Wilson, 2001*; *Eichenlaub et al., 2020*; *Dave and Margoliash, 2000*), for which a simple ordered chain is insufficient. To account for this latter phenomenon, our model takes elements from both chain and recurrent models in order to establish a new, hybrid framework for sequence learning. Our model's chain-like, modular structure enables it learn the order of elements, while the recurrent structure within that chain allows it to internally and flexibly learn those elements' duration. The structure of our network allows us to use a local, biophysically realistic learning rule to adjust the synaptic weights. A transient sequential input is presented during training, and over the course of training, our learning rule causes the weights to reach fixed points. After training, the network can then recall the uncompressed transient activity of the sequential input, complete with both duration and order, upon only partial reactivation (e.g. stimulating only the first element in the sequence).

## Results

### A model for sequence learning based on modular architecture and eligibility trace learning

Any model that can account for learning of behavioral temporal sequences must provide answers to a simple set of questions: what measures the timing of each event in a sequence? How is that information passed to the next, or the same computational unit? What mechanism provides the

appropriate delay? How are all of these elements, the order, duration, and specific timing, learned from experience? In a traditional 'chain-like' network with Hebbian learning, order of presented stimuli can be readily encoded by learning directional feed-forward connections between populations. However, this simple architecture proves insufficient for internally representing the duration of presented stimuli and their specific start and end times, as intrinsic time constants determine the speed at which the signal travels through the network (*Murray and Escola, 2017*; *Martinez et al., 2019*).

This issue is illustrated in *Figure 1*. In this schematic model, each module responds to a specific external stimulus. An external sequence activates these different modules in a given order and with externally determined durations for each element (*Figure 1a*). Learning can change the feed-forward synaptic efficacies between modules to reflect the order of the presented stimuli. Upon recall, triggered by activating the first stimulus, each of the encoded stimuli are activated at the correct order. However, the duration of activation within the module, as well as the timing of the activation of the

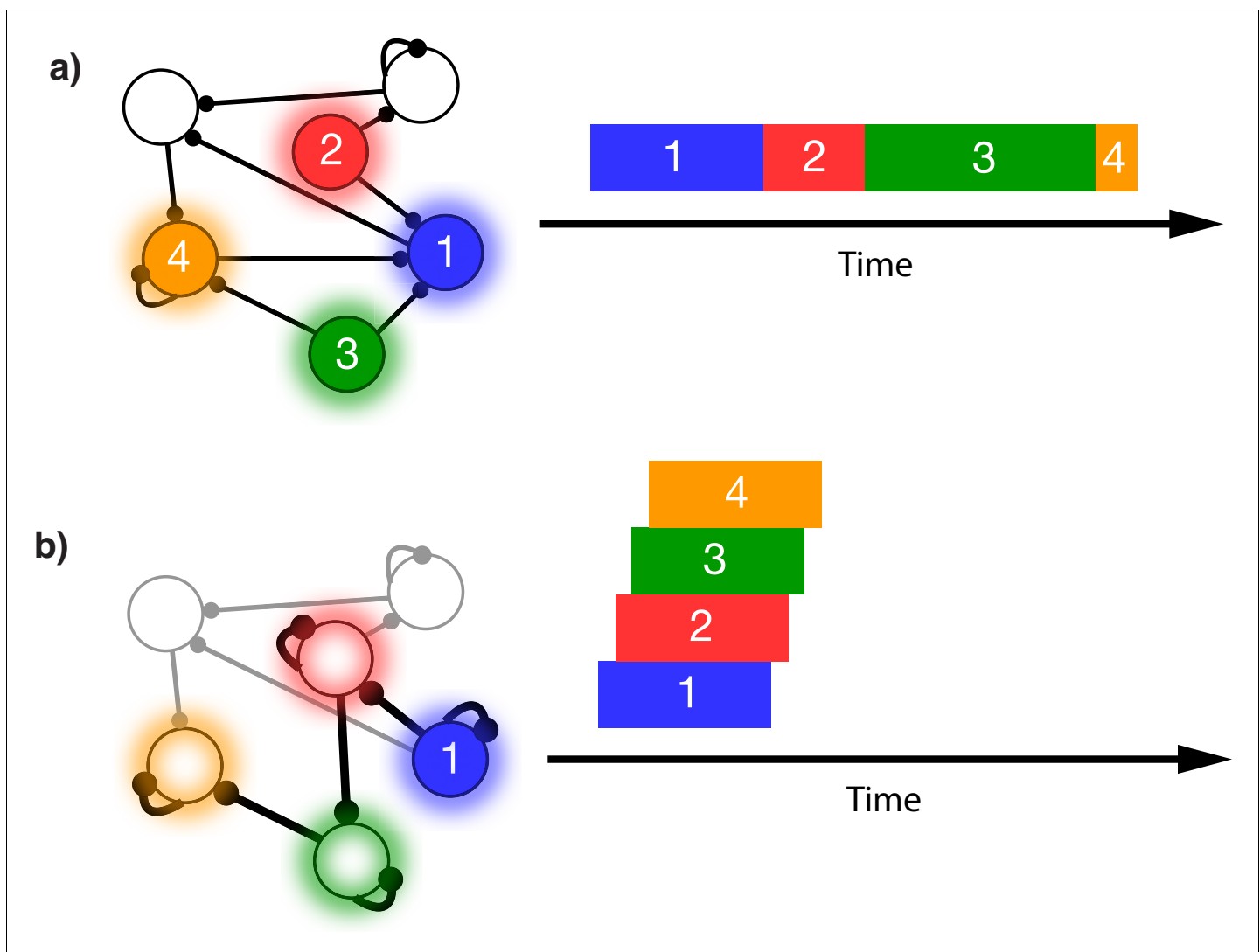

**Figure 1.** Sequence representation in networks. (a) A network composed of different populations of cells, each population is activated by a specific stimulus, and there are plastic connections between and within these populations. Initially these connections are random and weak. Upon presentation of a sequence of stimuli (filled circles, left), the populations will become activated for the duration and in the order in which they are stimulated (right). (b) After many presentations of a particular sequence, successful asymmetric Hebbian learning encodes the order of the stimuli into the synaptic weights of the network. After training, upon presentation of the first element of the sequence (filled circle, left), the network can recall (right) the order of presentation, but the timing and duration of each element is lost. In a generic network such as this, the timing of recall is determined by intrinsic time constants of the system and not the duration in the sequence that was presented.

subsequent module, is determined by the intrinsic time constants of the network (*Figure 1b*), causing a temporal compression of the recalled sequence. While certain time constants, asymmetrical inhibition, or adaptation with slow time constants could be manually placed in the network to facilitate recall of the particular sequence in *Figure 1a*, there does not yet exist a realistic, robust, and general solution to this problem.

In this work we present a network of spiking neurons that can learn the order, duration, and specific timing of sequence elements. This model uses local and biophysically realistic learning rules, in contrast to most RNN models (*DePasquale et al., 2018*; *Laje and Buonomano, 2013*; *Sussillo and Abbott, 2009*; *Nicola and Clopath, 2017*). As in the 'chain-like' models, the order is learned by modifying the feed-forward synaptic efficacies between modules. Unlike those models, the duration of each element is learned via modification of the recurrent connections within each module (*Fiete et al., 2010*; *Pereira and Brunel, 2019*; *Klos et al., 2018*). However, these two components are still not sufficient in order to avoid sequence compression during recall. In order to solve this problem, we assume additional structure within each module and in the allowed connections between modules. This additional structure allows us to avoid compression during recall while using relatively simple local learning rules. Consequently, the cellular response types generated by this network are consistent with experimental observation (*Shuler and Bear, 2006*; *Liu et al., 2015*), as described below.

Our network is composed of different modules that are selectively activated via feed-forward connections by different external stimuli. Within each module, there are two populations of excitatory cells as well as inhibitory cells (*Figure 2a*). The excitatory cells in both populations are identical in their intrinsic properties but differ in their learned and fixed connections with other cells within the module and in different modules. We name these two excitatory populations 'Timers' and 'Messengers' – the reason for these terms will become clear below as we describe their roles within the network. The 'Timer' cells learn strong recurrent connections with other Timer cells within the module. This strong recurrent connectivity results in long-lasting transient activity, which is used to represent the duration of a given stimuli. Previous studies have analyzed in detail the relationship between recurrent connectivity and duration of resulting transient activity following a stimulus (*Gavornik et al., 2009*; *Gavornik and Shouval, 2011*). Timers also excite the inhibitory cells and the 'Messenger' cells. Since the inhibitory cells receive input from the Timers, they have roughly the same temporal profile. However, inhibitory cells in the module decay slightly more quickly than their Timer counterparts, thanks to a combination of shorter time constants for synaptic activation (80 ms for excitatory, 10 ms for inhibitory), and small Timer to inhibitory weights (there are a number of degenerate sets of parameters which can facilitate quickly decaying inhibitory cells) (*Huertas et al., 2015*) (see 'Materials and methods' and Supplementary materials for more details). Owing to this temporal offset, Messenger cells selectively fire at the end of the Timers' transient activity, since they receive input from both the Timer cells and the (faster decaying) inhibitory cells. The temporally specific profile of the Messenger cells enables them to convey a temporally specific transition between elements, via learned feed-forward connections to Timer cells in other modules. The Timer/inhibitory/Messenger modular architecture can be constructed in a number of redundant ways, including simply via random distributions of connections (see previous work (*Huertas et al., 2015*) for a more detailed analysis).

For learning, we use a previously described reinforcement learning rule based on two competing, synapse-specific, Hebbian-activated eligibility traces (*Huertas et al., 2016*; *He et al., 2015*): one for long-term potentiation (LTP) and one for long-term depression (LTD) (see 'Materials and methods'). The presence of Hebbian activity at a given synapse activates the two traces at different rates (and to different saturation levels), and the absence of Hebbian activity causes the traces to decay at different rates. The change in synaptic weight is determined simply by the difference in these traces upon presentation of a reinforcement signal. We have assumed that on every transition between external stimuli, a 'novelty' signal causes a global release of a neuromodulator, which acts as a reinforcement signal (see 'Materials and methods'). The assumption of a temporally precise but spatially widespread neuromodulatory signal might seem at odds with common notions of temporally broad neuromodulator release, but they are indeed consistent with recent recordings in several neruomodulatory systems (*Howe and Dombeck, 2016*; *Hangya et al., 2015*). Eligibility traces for LTP have been found in multiple brain regions (*Yagishita et al., 2014*; *Bittner et al., 2017*; *Brzosko et al., 2017*), and eligibility traces for both LTP and LTD have been found in both visual and prefrontal

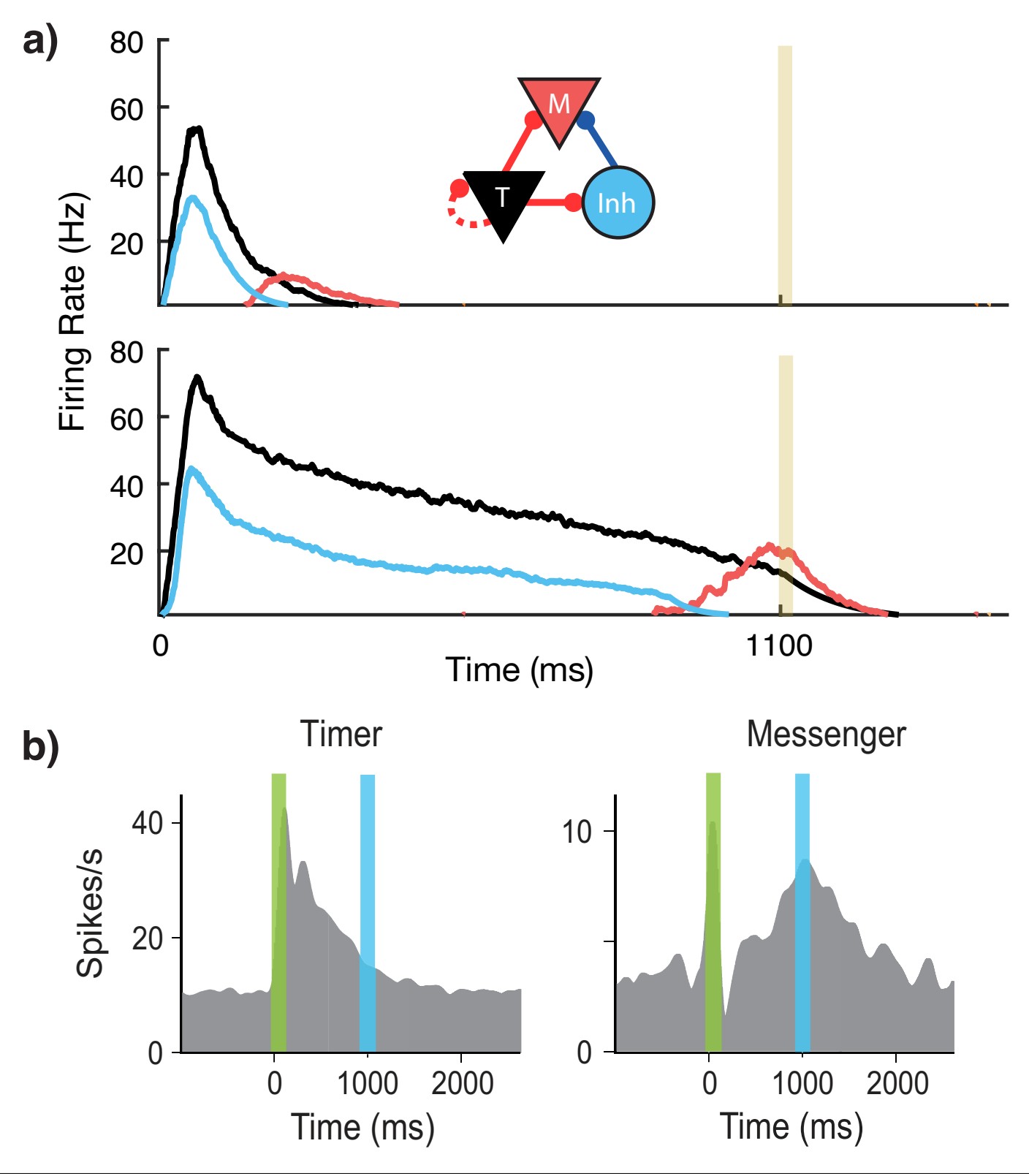

**Figure 2.** Microcircuit learns time intervals. (a) Mean firing rates of Timer (black), Messenger (red), and inhibitory populations (light blue) in a microcircuit before learning (top) and after learning (bottom) to represent an 1100 ms interval. Inset: Core neural architecture (CNA) microcircuit. Solid lines indicate fixed connections, while dotted lines indicate learned connections. (b) Timer and Messenger cell type responses to delayed reward task in V1. Green bar represents stimulus and blue bar represents reward. Schematic representation of data from *Liu et al. (2015)*.

cortex (**He et al., 2015**). We have used this rule because it can solve the temporal credit assignment problem, allowing the network to associate events distal in time, and because it reaches fixed points in both recurrent and feed-forward learning tasks (**Huertas et al., 2016**).

Using this rule and the described architecture within a single module, we find (**Figure 2a**) that the network naturally generates cells that learn to be active for the duration of the stimulus (Timers), and other cells that activate toward the end of the stimulus duration (Messengers). The Timer cells learn the duration of the stimulus via changes in the excitatory synaptic efficacies within the Timer population. The Messenger cells evolve their specific temporal profile due to the temporal profiles of their fixed excitatory and inhibitory inputs from the Timer cells and the inhibitory cells. These results are consistent with experimental observations in V1 circuits that learn the duration between a stimulus and a reward (**Shuler and Bear, 2006**; **Liu et al., 2015**; **Chubykin et al., 2013**), as shown in **Figure 2b**. These results replicate our previous results (**Huertas et al., 2015**) which used a learning rule with a single trace (**Gavornik et al., 2009**).

This modular microcircuit, in which both Timer and Messenger cells emerge from learning, acts as a 'core neural architecture' (CNA) (**Huertas et al., 2015**), an elemental package of basic temporal neural responses which can then be used within the larger network to create more complicated representations. This CNA functions as the basic microcircuit for our sequence model, and each 'element' of an input sequence is represented by one of these CNA circuits. One must note that the individual CNAs are plastic, as their temporal properties are not fixed but adapt to the environment, through recurrent learning in the Timer population. In a structure akin to the microcircuits found in cortical columns (**Potjans and Diesmann, 2014**), this model places distinct CNAs sensitive to particular visual stimuli in a columnar structure, as shown in **Figure 3a**. To prevent spurious excitation in our noisy, spiking network, there is soft winner-take-all (WTA) inhibition between the columns (see 'Materials and methods' and 'Discussion' for more details). When presenting a sequence of visual stimuli, different CNAs in turn become activated in sequence. Timer cells within those CNAs learn the duration of their particular stimuli via recurrent connections, while order is learned via feed-forward connections from Messengers in one column to Timers in a subsequently presented column. We will show that this modular architecture can overcome the problem of sequence compression encountered by 'chain-like' models.

## Learning and recalling the order and variable duration of presented sequences

Using the above described architecture and learning rule (see 'Materials and methods' for more details), the network is capable of robustly learning sequences of temporal intervals. **Figure 3** shows the network learning a sequence of four different elements, of duration 500, 1000, 700, and 1800 ms. The different stimuli here are labeled by color (blue, green, red, orange), and there are four corresponding columns in the network (out of 12) which are sensitive to these stimuli. In this example, the inputs are modeled after lateral geniculate nucleus (LGN) responses to spot stimuli (**Ruksenas et al., 2007**; **Mastronarde, 1987**, see **Figure 3—figure supplement 1**), but more generally one may consider these stimuli to be oriented gratings, specific natural images in a movie, or non-visual stimuli such as pitches of sounds in a song.

Before learning, presentation of the blue stimulus only produces a transient response in the CNA microcircuit housed in the blue-sensitive column (**Figure 3b**). During learning (**Figure 3c,d**), a sequence of stimuli is presented, and microcircuits in their respective columns learn to represent the duration of the stimulus which activates them. The different microcircuits also learn to 'chain' together in the order in which they were stimulated. After learning, presentation of the blue stimulus triggers a 'recall' of the entire sequence (**Figure 3e**): blue for 500 ms, red for 1000 ms, green for 700 ms, and orange for 1800 ms. The network is capable of learning sequences of temporal intervals where the individual elements can be anywhere from ~300 to ~1800 ms (see **Figure 3—figure supplement 2**) in duration, which agrees with the observed ranges used for training V1 circuits (**Gavornik and Bear, 2014**; **Shuler and Bear, 2006**). Additional examples of learned sequences are shown in **Figure 3—figure supplement 3**.

The network itself shows realistic spiking statistics, with interspike interval coefficients (ISIs) of variation near 1. **Figure 3—figure supplement 4** shows both spike rasters and ISIs for a single recall trial. The resulting firing rates are also roughly consistent with the experimentally observed Timers and Messengers (**Figure 2**).

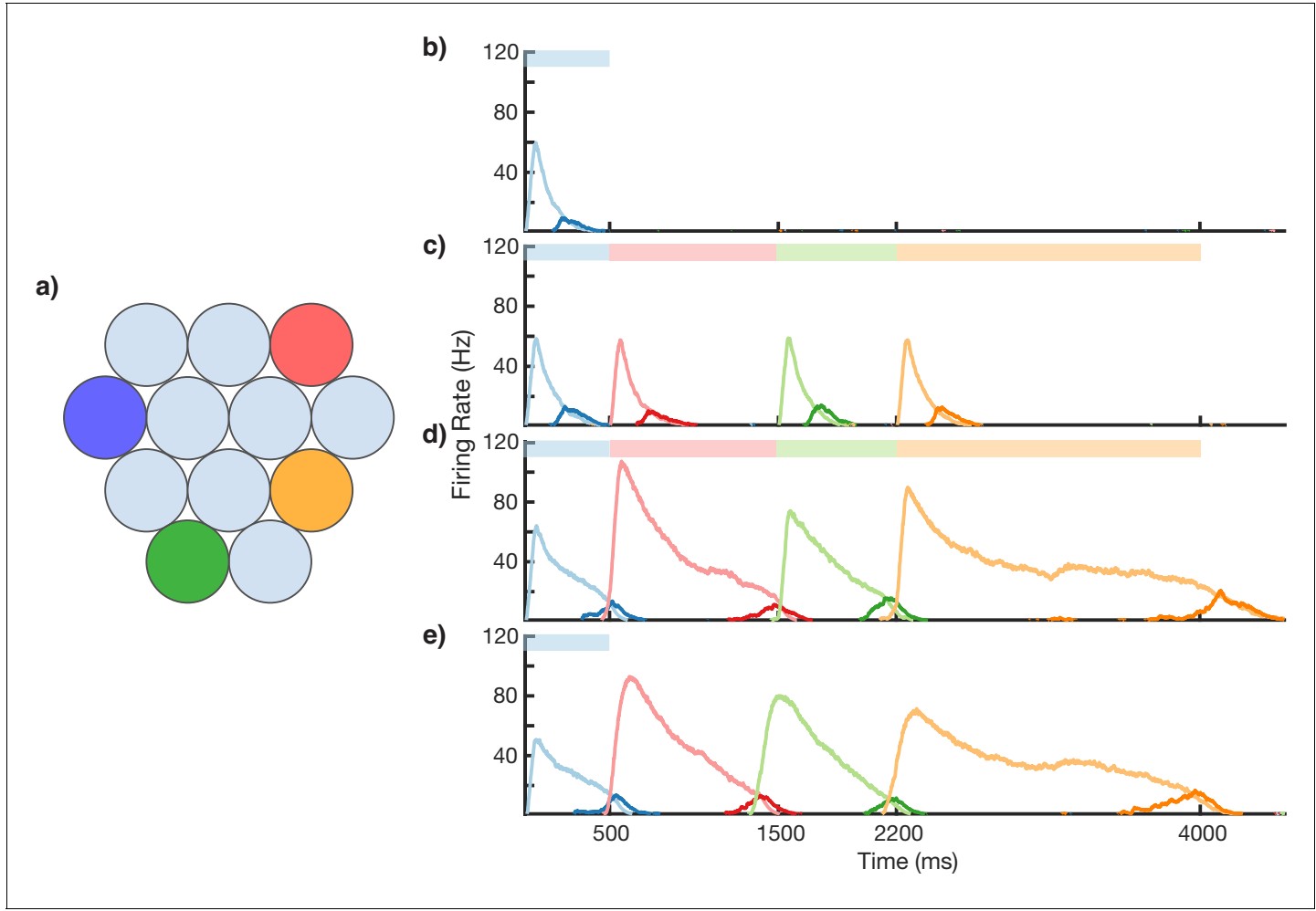

**Figure 3.** Sequence learning and recall. (**a**) Network of 12 columns, each containing a core neural architecture (CNA) microcircuit selective for a different stimulus. Columns containing microcircuits responding to blue, red, green, and orange stimuli are indicated. (**b–e**) Mean firing rates for Timer cells (light colors) and Messenger cells (dark colors) of four different columns during different stages of learning. Stimuli presented are shown as color bars in the top of plots. During learning, columns are stimulated in the sequence indicated by the color bars (500, 1000, 700, and 1800 ms for blue, red, green, and orange, respectively). (**b**) Before learning, the stimulation of a particular column only causes that column to be transiently active. (**c**) During the first trial of learning, all columns in the sequence become activated by the stimuli but have not yet learned to represent duration (through recurrent learning of Timer cells) or order (through feed-forward learning of the Messenger cells). (**d**) After many trials, the network learns to match the duration and order of presented stimuli. (**e**) After learning, presenting the first element in the sequence is sufficient for recall of the entire sequence. *Figure 3—figure supplements 1–6* provide additional information on the network's construction, accuracy, robustness, spiking statistics, dynamics, and limits. The online version of this article includes the following figure supplement(s) for figure 3:

**Figure supplement 1.** Input layer dynamics.

**Figure supplement 2.** Accuracy of learning and recall.

**Figure supplement 3.** Learning of different sequences.

**Figure supplement 4.** Spiking statistics in learned network.

**Figure supplement 5.** Eight element sequence recall.

**Figure supplement 6.** Rate-based learning and recall.

In simulations we have been able to learn a sequence of up to eight elements (*Figure 3—figure supplement 5*). The upper limit on the number of elements which can be learned in a sequence has not been fully explored due to the computational time required. For this work, sequences of four elements were chosen to match experimental results (*Gavornik and Bear, 2014*). The model presented here is a high dimensional spiking model, but similar results are obtained with a low dimensional rate model in which the activity of each population is presented by a single dynamical variable (for details, see 'Materials and methods' and *Figure 3—figure supplement 6*).

To gain an understanding of why learning succeeds in this model, we focus on the learning occurring between and within two selected modules in a sequence (*Figure 4*). Before training, presentation of an isolated stimulus only evokes a transient response of the CNA sensitive to that specific stimulus, since the Timer cells have yet to learn any recurrent connections, and the Messenger cells have yet to establish any feed-forward connections to the Timer cells in other columns (*Figure 4a*). During training, a particular sequence of inputs is presented and then repeated over many trials. Hebbian activity in the network triggers activation of synapse-specific LTP and LTD associated eligibility traces, which are then converted into changes in synaptic connections upon neuromodulator release (purple arrows in *Figure 4*), occurring here closely after a change in stimuli (see *Equations 8, 9, and 14* in 'Materials and methods'). Each trial pushes the weights in the network toward their fixed points (*Equations 17 and 18* in 'Materials and methods'). Hebbian activity within the Timer population causes activation of their respective eligibility traces, and subsequently an increase in their recurrent connections (*Figure 4—figure supplement 1*). As these lateral connections grow, Timer cells sustain their activity for longer, 'extending' their firing profile out in time toward the neuromodulator signal associated with the start of the subsequent stimulus. As this occurs, Messenger

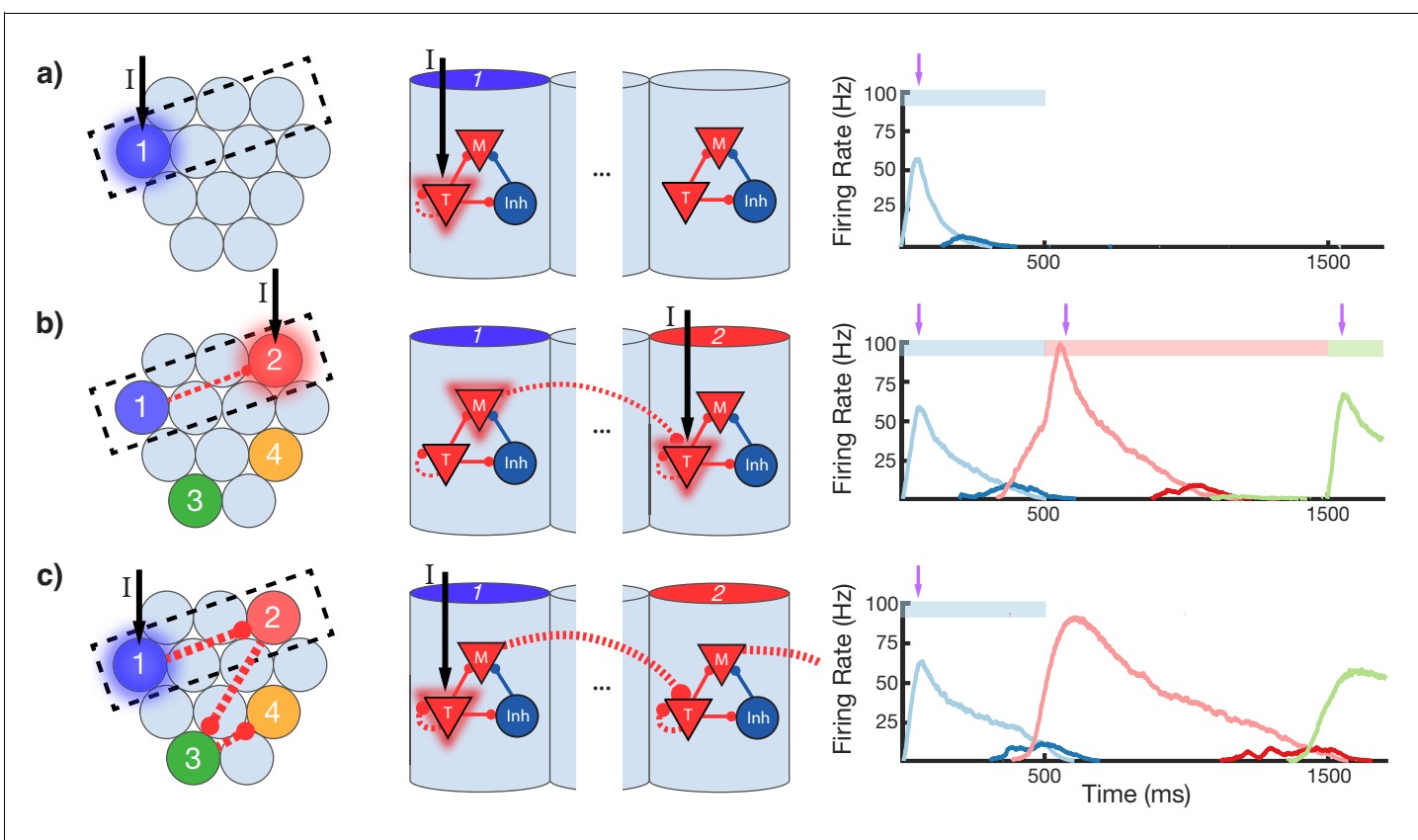

**Figure 4.** Change in connectivity patterns resulting from learning. (a) Before, (b) during, and (c) after learning a sequence. Left, view of columnar structure and learned intercolumnar connections. Dotted box indicates region shown in side view, middle. Middle, the detailed view of two columns and their core neural architectures (CNAs) and learned intracolumnar connectivity. Dotted lines indicate learned connections, continuous lines indicate fixed connections. Right, illustration of mean firing rates for color coded columns. Light colors indicate Timer cells, dark colors indicate Messenger cells. Color bars indicate stimulated columns. Purple arrows indicate global neuromodulator release. (a) Before learning, stimulus of a column's Timer (T) cells only causes that column to be transiently active. (b) If another column is stimulated shortly after the first, the Messenger (M) cells of the previous column will be coactive with the Timer cells of the stimulated column, thereby increasing the feed-forward synaptic weights between these two populations. (c) After learning, a physical synaptic pathway has been traced out which links columns in the temporal order in which they were stimulated during training. *Figure 4—figure supplements 1* and *2* demonstrate the dynamics of trace learning in the recurrent and feed-forward cases, respectively.

The online version of this article includes the following figure supplement(s) for figure 4:

**Figure supplement 1.** Recurrent learning evolution.

**Figure supplement 2.** Feed-forward learning evolution.

cells get 'dragged' along by the Timer cells, eventually coactivating with Timer cells of the column which is stimulated next (*Figure 4b*). This Hebbian coactivation triggers the eligibility traces of these feed-forward synaptic connections, before they too are converted into synaptic weight changes by the neuromodulator 'novelty' signal (*Figure 4—figure supplement 2*). After many trials (50–100 for the examples in this paper), weights in the network reach their steady-state values (see 'Materials and methods', *Figure 3—figure supplement 2*) and learning is complete. As the result of successful learning, a physical synaptic pathway has been traced out which encodes both the duration and order of the input. After learning, the encoded sequence can be recalled by stimulation of only the first element (*Figure 4c*). Importantly, recall does not demonstrate sequence compression, as the Messenger cells' activation (and thereby the activation of the next element) is appropriately temporally delayed.

Properly encoded durations and orders are the result of the fixed points in the learning rule, as described in the 'Materials and methods' section and in previous publications (*Gavornik and Shouval, 2011*; *Huertas et al., 2016*). Recurrent learning ends in a fixed point which sets the time $D$ between the end of firing in one column and the start of firing in the next (see *Equation 17*, *Figure 4—figure supplement 1*). Feed-forward learning results in a fixed point which determines the connection strength between Messenger and Timer cells in subsequent columns (see *Equation 18*, *Figure 4—figure supplement 2*). Formally, the fixed point sets the value of the Hebbian term, $H_{ij} = r_i \cdot r_j$, between the Messenger cells and the Timer cells in the next column at the time of reward, and implicitly this results in setting the connection strengths $W_{ij}$. Both these fixed points depend on the parameters of the learning rule, which can be chosen such that these terms achieve desired values ($D$ arbitrarily small, $H_{ij}$ to a fixed value at time of reward). Such learning can then correctly encode any presented sequence. Earlier work examines in detail the dependence of $D$ on network parameters (*Gavornik and Shouval, 2011*; *Huertas et al., 2016*; *He et al., 2015*).

Empirically, temporal accuracy of recall depends on many non-trivial factors (i.e. length of individual elements, length of entire sequence, placement of short elements near long elements, etc.), owing to the many non-trivial effects of stochasticity of the spiking network (spike rasters are shown in *Figure 3—figure supplement 4*). In addition to fluctuations in recall accuracy, there can also be fluctuations in learning accuracy, as randomness in spiking can happen to accumulate such that the traces (and therefore the fixed points) are also sufficiently modified over the course of training. Over the whole network, over the course of many trials, and over the course of learning instances, these effects tend to wash out. Reported times in the recalled sequence generally match the times of the input sequence to within 10% (see *Figure 3—figure supplement 2*). This model does not observe any intrinsic bias toward over- or under-predicting, as this can be modulated by network parameters and can change stochastically from trial to trial or from element to element.

Our model also exhibits robustness to changes in its parameters. In *Figure 5*, we demonstrate the network successfully recalling a learned four element sequence, in which the fixed connections that establish the Timer and Messenger cells are modified by +/- 20%. Furthermore, the mean reported time in a two-column network is conserved after application of random fluctuations to the learning parameters (*Figure 5—figure supplement 1*). In addition, the network can also function if the synaptic time constant for excitatory connection is shortened from 80 to 20 ms (*Figure 5—figure supplement 2*). A long time constant of excitatory connections is often used in models of working memory (*Lisman et al., 1998*) and models of long-lasting but transient recurrent networks (*Gavornik et al., 2009*; *Gavornik and Shouval, 2011*). Such long time constants make the network mode robust, able to learn longer transient durations and to operate in a more physiological range (*Wang, 2001*), and there is evidence for such time constants in some brain regions (*Wang et al., 2013*). Nevertheless, our network can operate with faster excitatory time constants (*Figure 5—figure supplement 2*), but shorter synaptic time constants limit the duration of elements that can be learned.

While the results shown in this work were obtained using a two-trace learning (TTL) rule, the network can also be trained with a learning rule based on a single trace (*Gavornik et al., 2009*; *Gavornik and Shouval, 2011*), and the results are similar to those demonstrated here (one-trace results not shown).

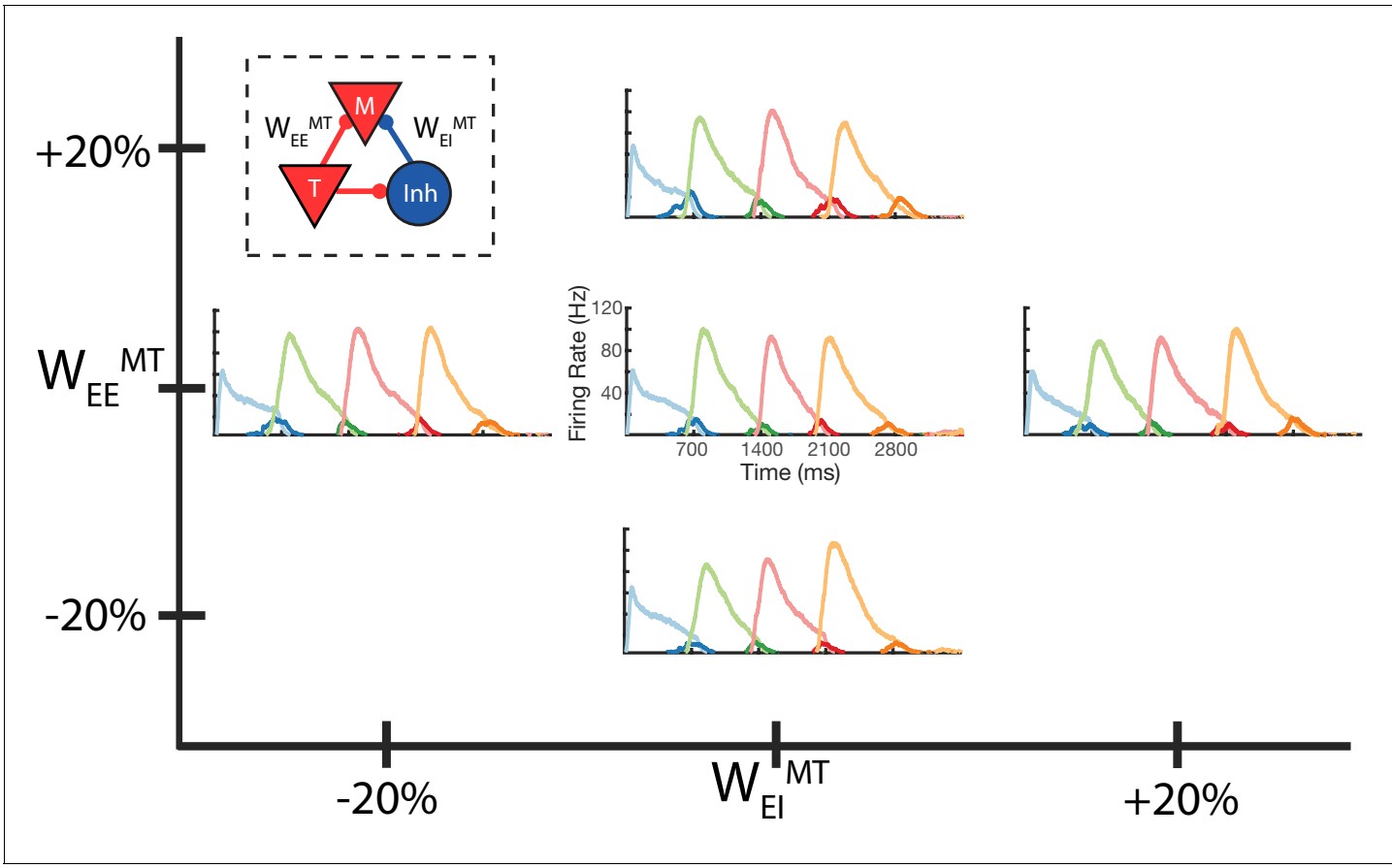

**Figure 5.** Robustness to core neural architecture (CNA) weight changes. Firing rates of four columns, after learning a four element sequence, each of 700 ms duration. Only the first element is stimulated for recall. Before learning, static CNA weights $W_{EE}^{MT}$ and $W_{EI}^{MT}$ are either set as in *Supplementary file 1* (center plot), or independently adjusted +/− 20% from their values in *Supplementary file 1*. In each of the four 'modified' cases, sequence learning is still successful and retains the correct timing. The most noticeable difference is along the $W_{EE}^{MT}$ axis, where the amplitude of the Messenger cells can be seen to increase as $W_{EE}^{MT}$ increases. Inset: CNA microcircuit with labeled connections. *Figure 5—figure supplement 1* demonstrates a two-column network's robustness to random variations in the learning parameters. *Figure 5—figure supplement 2* shows the success and failure cases for a network with a 20 ms excitatory time constant.

The online version of this article includes the following figure supplement(s) for figure 5:

**Figure supplement 1.** Robustness of parameter randomization.

**Figure supplement 2.** Sequence learning with 20 ms excitatory time constant.

## Learning and recalling non-Markovian sequences

In the modular network described above, the transitions from each module to the next depend only on the identity of the current module that is active; such a model is formally called a Markovian model (*Gillespie, 1991*). While the Markov property is typically discussed in the context of probabilistic models, here we apply it to deterministic sequences as well. A Markovian model can only reproduce specific types of sequences and is unable to reproduce a sequence in which the same element is repeated more than once and is followed each time by a different element, for example, the sequence ABAC. The columnar network described above is essentially Markovian since activation of a neural population will necessarily feed forward to all other populations it is connected to, no matter the history or context. Behaviorally, learning of non-Markovian sequences is ubiquitous, and cells responsible for producing and learning non-Markovian sequences exhibit non-Markovian activation (*Cohen et al., 2020*).

To learn non-Markovian sequences, we modify the network structure while maintaining local learning rules. Ideally, the network should be able to learn sequences which are in themselves non-

Markovian (ABACAD), as well as simultaneous combinations of sequences which are non-Markovian when learned together (ACE and BCD). We will demonstrate both cases here.

For the network to learn and recall non-Markovian sequences, it must somehow keep track of its history and use this history to inform transitions during sequence learning and recall. To this end, we include two additional stages to the network (*Figure 6*). The first is a fixed (non-learning) recurrent network, sometimes called a 'reservoir' (as in reservoir computing) or a 'liquid' (as in a liquid state machine) ( *Maass et al., 2002*; *Maass et al., 2004*), which receives inputs from the Messenger cells in the main columnar network. Owing to these inputs and due to its strong recurrent connectivity, the current state of the reservoir network is highly dependent on the history of network activity.

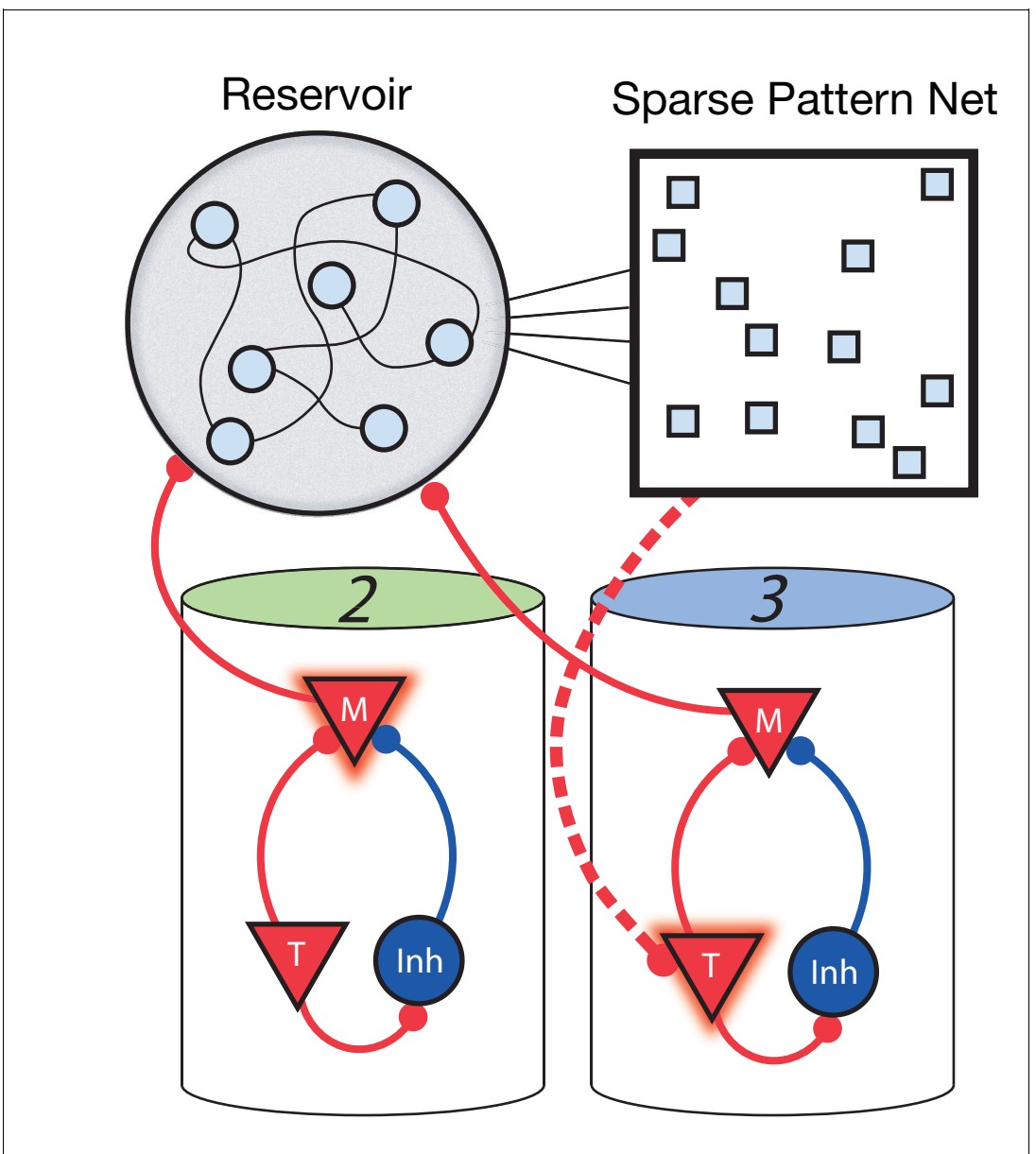

**Figure 6.** Non-Markovian sequence learning and recall. Three-stage network. Two sequentially activated columns (2–3) learn to connect to each other through a reservoir and sparse pattern net. At time *t*, Messenger cells from column 2 are active and act as inputs into the reservoir (earlier, Messenger cells from column 1 also fed into the reservoir). The sparse pattern net receives input from the reservoir, so as to be a unique representation of the history of the network up to and including time *t*. Timer cells active at *t* + Δ*t* (column 3) connect to the sparse pattern via Hebbian learning.

Therefore, it acts as a long-term memory of the state of the columnar network. The second additional stage is a high dimensional, sparse, non-linear network which receives input from the reservoir, serving to project the reservoir states into a space where they are highly separated and non-overlapping. The result is that a given pattern in this sparse network at time $t$ uniquely identifies the history of the main network up to and including time $t$. Since these patterns are highly non-overlapping (due to the sparsity and non-linearity), a particular pattern at time $t$ can use simple, local, and biophysically realistic Hebbian learning to connect to Timer cells firing at time $t + \Delta t$ in the main network (direct Messenger to Timer feed-forward learning is removed in this non-Markovian example).

We test the ability of this three-stage network to learn long, non-Markovian sequences with repeated elements by presenting a sequence blue-green-blue-red-blue-orange (BGBRBO) during training (*Figure 7*). A simplified, rate-based version of the columnar network is used to reduce computation time (see 'Materials and methods). The presented sequence is such that there are three different transitions from blue to another element, and each stimulus is also presented for a different duration, making this a non-trivial problem. Before learning (*Figure 7a*), the blue stimulus does not evoke any recall, only triggering a transient response of the blue-responsive column. During learning (*Figure 7b,c*), Timer cells learn the duration of their stimulus, as before, but now it is the sparse pattern net which is using Hebbian learning to feed forward to Timers in subsequently stimulated

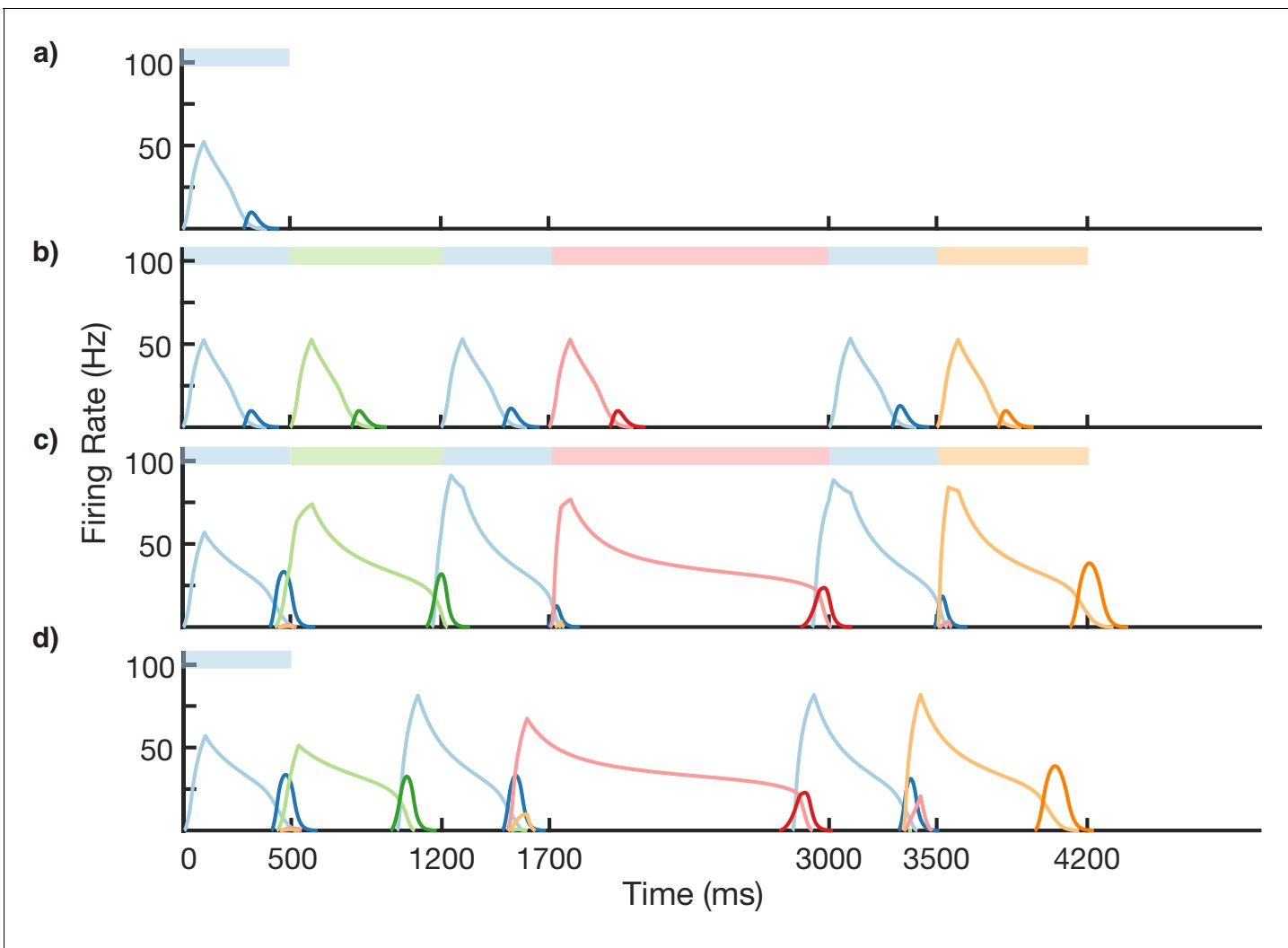

**Figure 7.** Non-Markovian sequence learning and recall. Mean firing rates for Timer cells (light colors) and Messenger cells (dark colors) of four different columns during different stages of learning (before, first trial of learning, last trial of learning, after learning). Stimuli presented are shown in color bars inset in top of plots (500, 700, 500, 1300, 500, and 700 ms for blue, green, blue, red, blue, and orange, respectively).

columns. After learning (*Figure 7d*), external input to the blue stimulus is sufficient to trigger recall of the entire trained non-Markovian sequence. In this case, each blue element in the sequence has the same duration; this is owing to the fact that our TTL rule currently only supports one dynamic attractor per population. In order for repeated elements to have different durations during recall, an appropriate learning rule must be capable of creating multiple attractors within that element's Timer population, with each attractor triggering a different duration of activity.

We also demonstrate an example of simultaneous learning of two sequences with a shared element (*Figure 8*). First, a sequence blue-red-orange (BRO) is trained, and then a sequence green-red-purple (GRP) is also learned. Each of the elements in these sequences has a duration of 500 ms. With both these sequences learned and stored simultaneously in the same network, transitions from red are non-Markovian – red should transition to orange if it was preceded by blue and should transition to purple if it was preceded by green. *Figure 8a* shows recall upon stimulation of blue, while *Figure 8b* shows recall upon stimulation of green. In both cases, the sequence makes the correct transition from red to the appropriate third element. Recall of both sequences is also robust to perturbations in the initial state of the reservoir, as shown in (*Figure 8—figure supplement 1*.)

Of note in these two simulations is that the 'incorrect' transitions do get slightly activated, as the sparse patterns responsible for these transitions are not completely non-overlapping. There is an interplay here between the robustness and uniqueness of these patterns – the sparser they are, the

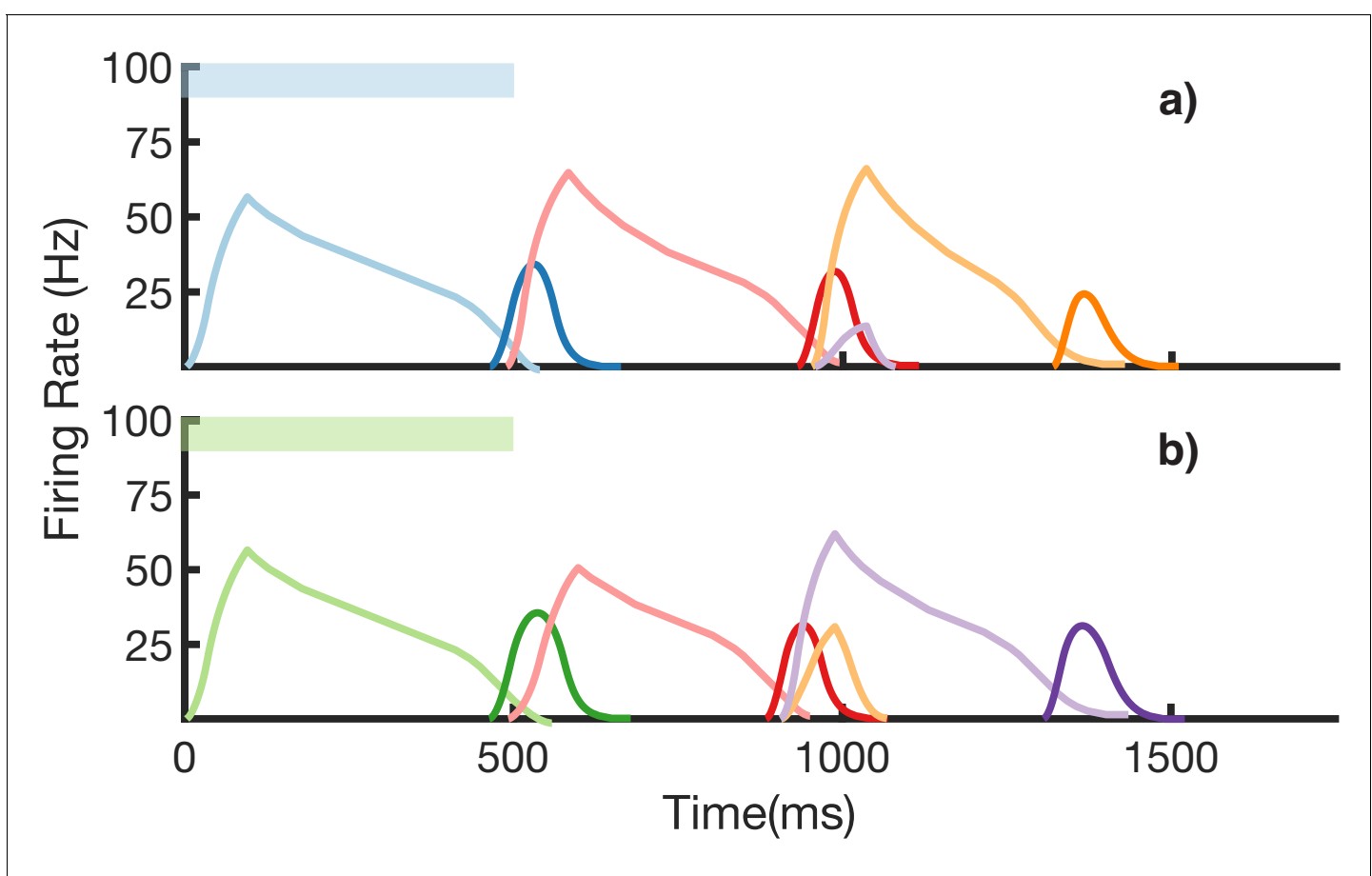

**Figure 8.** Recall of two overlapping sequences. Mean firing rates for Timer cells (light colors) and Messenger cells (dark colors) during recall of two sequences. Both blue-red-orange (BRO) and green-red-purple (GRP) have been stored in the network via learning. (**a**) Recall of BRO, following presentation of a blue stimulus. (**b**) Recall of GRP, following presentation of a green stimulus. Note that R transitions to a different element in the two sequences. *Figure 8—figure supplement 1* demonstrates the network's robustness to perturbations of the reservoir's initial state.

The online version of this article includes the following figure supplement(s) for figure 8:

**Figure supplement 1.** Robustness in non-Markovian recall.

more non-overlapping they are, but also the more sensitive they are to noise. Other parameters in the model, such as the level of recurrence in the reservoir, or the projection strengths from the reservoir to the sparse net, also affect the robustness/uniqueness of the patterns. For the simulations shown, a set of parameters (*Supplementary file 2*) was empirically chosen so that a handful of non-Markovian transitions could be reliably encoded. Though we have not optimized the network parameters, or fully characterized its properties, these results demonstrate that such an architecture can overcome the problem of non-Markovian sequences.

The addition of the reservoir and sparse network is essential for the ability to encode and replay non-Markovian sequences, but they alone are not sufficient for sequence learning and recall in our model. One must note that the reservoir receives its input from the modular network, not from the environment, and therefore if the modular network does not learn, the inputs to the reservoir would not be generated appropriately. In this formulation, during learning, the entire sequence is presented as inputs, but to trigger recall after learning, only the first element of the learned sequence is presented. In short, the input during learning and recall is different. Typical reservoir computing models can learn an output, given a specific input signal, but if that input signal changes, then so would the resulting output. This means that typically the reservoir alone cannot learn to recall with only the first element of the sequence as input, given that it was presented the whole sequence during training.

We use rate-based versions of all three stages for simplicity and computational efficiency, but a spiking based model is likely to have similar results. In our Markovian modular model described above, we use a fully spiking model. The rate-based approximation, which we use here in the non-Markovian case, is shown to produce similar results (*Figure 3—figure supplement 6*). The reservoir itself does not learn and needs only to be highly recurrent to recapture the demonstrated functionality. Therefore, a spiking implementation is likely to work, albeit using many more neurons and at a much higher computational cost. The sparse network as used here is binary (see 'Materials and methods'), but non-binary sparse representations are likely to produce similar results. Transitioning all three stages to using actual spiking neurons would require much time, computational power, and parameter adjustments, but there are no clear fundamental roadblocks to doing so.

We have chosen this simple sparse representation of this three-stage network, not because it is a biophysically realistic implementation but in order to demonstrate the concept that such an addition is sufficient for learning and expressing non-Markovian sequences, while still using local learning rules. The viability of this relatively simple, compartmentalized structure demonstrates two things: first, that our original columnar network can be used as a building block in more complicated tasks, and second, that even a complex task such as learning non-Markovian sequences does not require non-local learning rules to perform. There are, in general, likely a large number of architectures which would effectively achieve a similar aim – take a very complex problem like non-Markovian sequence learning and constrain the possible solution space, allowing for a local learning rule to 'finish the job'. Other groups have approached non-Markovian sequences by including additional hierarchy or long synaptic time constants (*Hawkins and Ahmad, 2016*; *Tully et al., 2016*), and solutions like RNNs handle such sequences natively (albeit with non-local learning rules) since they are continuous systems with a dependence on a relatively long history when compared to the intrinsic time constants. We combine these two methods, using highly recurrent networks in the context of a larger architecture, and this combination allows us to maintain local and biophysically realistic learning rules.

## Discussion

In this work, we demonstrate the ability of a modular, spiking network to use local, biophysically realistic learning rules to learn and recall sequences, correctly reporting the duration and order of the individual elements. In combining modular, heterogenous structure with a learning rule based on eligibility traces, the model can accurately learn and recall sequences of up to at least eight elements, with each element anywhere from ~300 to ~1800 ms in duration. We have also shown a modified architecture that is capable of learning and recalling non-Markovian sequences with multiple history-dependent transitions.

The capabilities, construction, and rules of our model are significantly different from most contemporary and historical models of sequence learning, such as synfire chains and RNNs. In particular, we are not aware of another model that uses spiking networks and local, biophysically realistic learning rules, along with experimentally observed cell responses, to robustly learn both the order and duration of presented sequences of stimuli. Previous approaches which have attempted to internally learn both duration and order of elements (*Veliz-Cuba et al., 2015*) have been highly sensitive to changes in parameters and/or were not based on experimentally observed cell responses. Other types of hybrid models combine chain structures with additional hierarchy/functionality (*Murray and Escola, 2017*; *Martinez et al., 2019*), but either requires manually set adaptation time constants that are specific for a set duration to represent the duration of elements or do not treat the duration of elements as variable at all. Recently another model has been proposed, based on a periodic and essentially deterministic timing network, which also uses biologically plausible learning rules to learn to represent single non-Markovian sequences (*Maes et al., 2020*). However, it cannot learn arbitrarily long (and self-terminating) sequences, such as those presented in (*Figure 7*), nor can it learn several different sequences and replay them separately, such as those presented in (*Figure 8*)

The modified three-stage model is capable of learning non-Markovian sequences because of inclusion of a non-learning recurrent network that stores memory over longer durations than the modular network. However, the model's ability to generate learned sequences depends crucially on the backbone of the modular network – because of this, we can use a non-plastic, highly recurrent reservoir and maintain biophysically realistic learning rules in the network. This modified model also differs in functionality from traditional RNNs, as in our model, the stimulus presented during training (the entire sequence) is different from the stimulus presented after learning in order to trigger recall (usually the first element is sufficient). In general, the number of elements necessary to trigger recall is the same as the number of elements needed to disambiguate the sequence (i.e. 'A' would be sufficient to trigger ABCD in a network embedded with learned sequences ABCD and EFGH, but 'AB' would be required to trigger ABCD in a network embedded with learned sequences ABCD and AFGH). In a way, our network acts as an autoencoder, both learning a reduced encoding (e.g. 'A' instead of 'ABCD') and reconstructing the original external input from this reduced encoding.

The reservoir and sparse network components of our three-stage model could be thought to arise from a projection from other cortical or subcortical areas. Functionally similar networks (ones that take complex, multimodal, and dynamic context and repackage it into sparse, separated patterns) have been observed in the dentate gyrus (*Leutgeb et al., 2007*; *van Dijk and Fenton, 2018*) and the cerebellum (*Chadderton et al., 2004*; *Billings et al., 2014*). However, these model components could also be thought of as part of the same cortical network, partially segregated in function but not necessarily by location. Pattern separation is likely a common neural computation that might occur in many brain areas, so we make no particular predictions about the locations of these network components.

Our model suggests that the types of single cell responses in V1 observed in delayed reward tasks (Timers and Messengers) (*Shuler and Bear, 2006*; *Liu et al., 2015*; *Chubykin et al., 2013*) will also be present when learning a visual sequence. Furthermore, we predict that distinct populations of these cells are sensitive to and can learn the dynamics of distinct stimuli. This functional modularity could be physically implemented as physically compartmentalized in columns and layers, though this is not strictly required. We present here two equivalent possible configurations of the network. The first, *Figure 9a*, shows an architecture that more directly implements the CNA, where inhibitory neurons are physically located next to their associated excitation for clarity, and long-range inhibitory connections exist. *Figure 9b* displays a more realistic structure by restricting inhibitory connections to be local. Both architectures are functionally identical, and there are a number of other redundant constructions which recapture the behavior described in this work.

Recent results showing sequence learning and recall in visual cortex support the idea that these cell types are implicated in sequence learning (*Gavornik and Bear, 2014*). When multiple visual stimuli are presented sequentially, Local Field Potential (LFP) recordings indicate Timer-like responses (long, sustained potentiation) in layer V, with Messenger-like responses (short bursts centered around the transition between elements) arising in layer II/III (see Figure 4 from *Gavornik and Bear, 2014*). These results are consistent with the hypothesis that Timers and Messengers are indeed compartmentalized into different cortical layers.

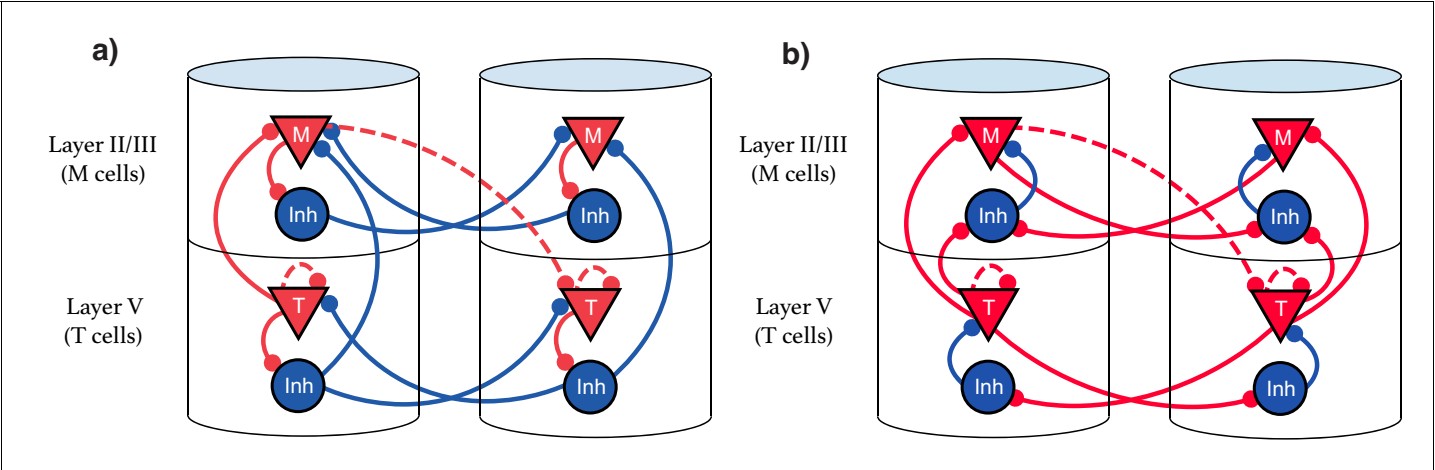

**Figure 9.** Explicit microcircuit structure. (**a,b**) Two examples of complete microcircuit structure displayed in laminar architecture of cortical columns. Dashed lines represented learned connections, while continuous lines represent fixed connections. (**a**) Intercolumn inhibition produces soft winner-take-all dynamics between columns, interlaminar inhibition generates core neural architecture (CNA). (**b**) Same functionality as (**a**) but rearranged so as to only have local inhibition.

Our model makes several testable predictions. It predicts that: (1) After learning a sequence, Messenger cells will more strongly functionally connect to Timer cells that represent subsequent stimuli, than to Timers which represent previous stimuli, or to other Messengers. (2) Learning sequences with long duration elements will increase lateral connection efficacies between Timers within the same modules. However, learning sequences with short duration elements may actually weaken those same lateral connections, depending on initial conditions. (3) There will be a population of inhibitory cells within each module that have firing properties similar to Timers but that decay more quickly (*Figure 2a*).

From a theoretical perspective, the main takeaway is that the architecture of our model enables it to accurately learn and recall sequences while maintaining local learning rules. If we were to start with a homogenous, randomly connected pool of neurons, sequence learning would require precise credit assignment, which is very difficult or impossible to calculate locally. Instead, the modular structure and the functionally heterogeneous cell types of the CNA perform some of this credit assignment implicitly, by breaking down a complex, difficult to learn task into a hierarchy of well-defined and easy to learn tasks. The Timers learn the duration, and the Messengers learn the order (components in the non-Markovian formulation have similarly simple and segregated tasks). While these restrictions end up limiting the types of outputs our network can represent, they enable the use of local learning rules.

## Materials and methods

### Network architecture

In a totally unstructured network, the learning rule described below would be insufficient for the task of sequence learning and memory. In exchange for the freedom of online, biophysically realistic reinforcement learning, we must presuppose some restrictions on the macro structure of the network. The structure imposed on the network has two components: a modular columnar structure with restricted connections and the weight distribution of the non-plastic synaptic efficacies. The weight distribution results in the emergence of distinct Timer and Messenger cell types, and the columnar structure places populations of these cells into stimulus-specific modules.

The network consists of 12 different 'columns', each containing 100 excitatory Timer cells, 100 inhibitory Timer cells, 100 excitatory Messenger cells, and 100 inhibitory Messenger cells, all assembled in a CNA microcircuit. Neither the particular number nor the ratio of the cell types is a strict requirement. These populations can be represented by many spiking neurons, few spiking neurons, or a single mean field neuron, and the populations can emerge even from distributions of random

connections (*Huertas et al., 2015*). Each column is tuned to be selective for a particular stimulus from the input layer, such that a sequence of external stimuli (e.g. ABCD or 'red-green-orange-blue') will trigger a corresponding sequence of input-selective columns.

To simulate presentation of visual cues, the network is stimulated by an input layer designed to mimic spiking inputs from the LGN. Each stimulus is represented in the input layer by a Poisson spike train with a 50 ms pulsed peak, in accordance with the observed cellular responses in LGN (*Ruksenas et al., 2007*; *Mastronarde, 1987*). In general, this spike train can also include a decaying exponential tail, but short pulses were chosen both for analytical simplicity and to match experimental data (*Ruksenas et al., 2007*; *Mastronarde, 1987*), as shown in *Figure 3—figure supplement 1*. The cells of the input layer feed directly into the respective Timer cells of the columnar network.

Intercolumnar inhibition exists between Timer populations and between Messenger populations in the different columns, for the purposes of soft WTA dynamics between the columns of the network. This WTA inhibition can be thought of functionally as long-range inhibitory connections (*Figure 8a*), but in reality, is much more likely to be long-range excitatory connections synapsing onto local inhibition (*Figure 8b*). Such inhibition is generally not necessary in the low-noise case (or in a rate-based model), but the stochastic, spiking nature of the network makes it a desirable practical inclusion to guard against spurious runaway excitatory chains.

For the purposes of this paper, Timer cells can only learn to connect to other Timer cells within their column, and Messenger cells can only learn to connect to (any) Timer cells outside of their column. In case of the extended network that can learn non-Markov sequences, Messenger cells connect to the reservoir and not directly to Timer cells in other columns, as described above. While we include these restrictions for practical purposes, previous publications have shown all possible intracolumnar excitatory connections to be learnable with the use of a single trace eligibility rule (*Huertas et al., 2015*). The strengths of non-plastic connections within columns are modeled after those learned in previous publications (*Huertas et al., 2015*). These fixed connections serve to establish the Timer and Messenger cells used in the network. In *Figure 5*, we demonstrate that successful sequence learning is robust to +/− 20% changes in these connections. By only learning task relevant connections, we simplify the computation and analysis while maintaining the complete general functionality of variable sequence learning and recall.

## Spiking and rate-based dynamics

The network comprises microcircuits of both excitatory and inhibitory spiking leaky-integrate-and-fire neurons placed in a modular architecture akin to that observed in cortical columns. The following equations describe the membrane dynamics for each model neuron i:

$$C\frac{dv_i}{dt} = g_L(E_L - v_i)g_{E,i}(E_E - v_i)g_{I,i}(E_I - v_i)\sigma \tag{1}$$

$$\frac{ds_i}{dt} = -\frac{s_i}{\tau_s}\rho(1 - s_i)\sum_k \delta(t - t_k^i) \tag{2}$$

Here, subscripts L, E, and I refer to leak, excitatory, and inhibitory, respectively. g refers to the corresponding conductance, and E to the corresponding reversal potentials. $v_i$ and $s_i$ are the membrane potential and synaptic activation, respectively, of neuron i. $\sigma$ is a random noise term. Once the membrane potential reaches a threshold $v_{th}$, the neuron spikes and enters a refractory period $t_{ref}$. Each spike (at time $t_k^i$) updates the synaptic activation $s_i$ by an amount $\rho(1 - s_i)$, and in the absence of spikes, synaptic activation decays exponentially. Conductance $g$ is the product of the synaptic weight matrix with the synaptic activation, summed over all presynaptic neurons:

$$g_{\alpha,i} = \sum_j W_{ij}^\alpha s_j \tag{3}$$

Here, $\alpha$ can be either $E$ (excitatory) or $I$ (inhibitory), and $W_{ij}^\alpha$ are the connection strengths from neuron *j* to neuron *i*. A firing rate estimate for each neuron $r_i$ is calculated by an exponential filter of the spikes at times $t_k^i$, with a time constant $\tau_r$.

$$\tau_r \frac{dr_i}{dt} = -r_i \sum_k \delta\left(t - t_k^i\right) \tag{4}$$

Our excitatory time constant is notably long (80 ms), but this is not strictly required for our model. This is shown in *Figure 5—figure supplement 2*, in which we set the excitatory time constant to 20 ms and demonstrate successful learning of a sequence with elements 500 ms long each. However, the ability of the Timers to learn long times via their recurrent connections (without prohibitively small learning rates) depends on such large time constants, which are common in working memory literature (*Gavornik and Shouval, 2011*; *Lisman et al., 1998*; *Wang et al., 2013*). *Figure 5—figure supplement 2b* shows that the Timer cells reach bistability when trying to learn 1000 ms with a 20 ms time constant, causing failure in learning. The relationship between reported time, recurrent weights, and time constants in Timer-like cells is analyzed in detail in previous work (*Gavornik et al., 2009*; *Gavornik and Shouval, 2011*; *Huertas et al., 2016*). Although there is some evidence for a slow time constant in prefornntal cortex (PFC) (*Wang et al., 2013*), this might not be so in sensory cortex. There are alternative ways to acquire a slow time constant that can facilitate learning of long interval times, such as derivative feedback (*Lim and Goldman, 2013*) or active intrinsic conductance (*Gavornik and Shouval, 2011*; *Fransén et al., 2006*).

For the rate-based version of these dynamics (used in the three-stage network model), each population of spiking neurons is represented by a single rate-based unit, which is governed by the following equations:

$$\tau_u \frac{du_i}{dt} = -u_i + \sum_{j,\alpha} W_{ij}^\alpha \xi(u_j) \tag{5}$$

$$r_i = \xi(u_i) \tag{6}$$

where $\tau_u$ is the characteristic time constant and $r_i$ is the resulting firing rate for neuron $i$. $\xi$ is a piecewise non-linear transfer function used in previous sequential activation models (*Brunel, 2003*):

$$\xi(u) = \begin{cases} 0 \ if \ u \leq \theta \\ v\frac{u-\theta}{(u_c-\theta)^2} \ if \ \theta < u < u_c \\ 2v\sqrt{\frac{u-\theta}{u_c-\theta} - \frac{3}{4}} \ if \ u_c < u \end{cases} \tag{7}$$

where $u_c$ is the critical activity level, $\theta$ is a lower threshold, and $v$ a scaling parameter.

## TTL rule

In place of a pair-based spike timing-dependent rule or rate-based Hebbian rule, which fails to solve the temporal credit assignment problem, the network learns based on 'eligibility traces' for both LTP and LTD (*Huertas et al., 2015*; *Frémaux and Gerstner, 2015*). These eligibility traces are synapse-specific markers that are activated via a Hebbian coincidence of activity between the pre- and postsynaptic cells. At a maximally allowed activation level, these traces saturate, and in the absence of Hebbian activity, these traces decay. LTP and LTD traces are distinct in that they activate, decay, and saturate at different rates/levels. These dynamics are described in the following equations:

$$\tau^p \frac{dT_{ij}^p}{dt} = -T_{ij}^p + \eta^p H_{ij}\left(T_{max}^p - T_{ij}^p\right) \tag{8}$$

$$\tau^d \frac{dT_{ij}^d}{dt} = -T_{ij}^d + \eta^d H_{ij}\left(T_{max}^d - T_{ij}^d\right) \tag{9}$$

The superscripts $p$ and $d$ indicate LTP or LTD synaptic eligibility traces, respectively. Here, $T_{ij}^a$ (where $a \in (p,d)$) is the eligibility trace located at the synapse between the $j$th presynaptic cell and the $i$th postsynaptic cell. The Hebbian activity, $H_{ij}$, is a simple multiplication $r_i \cdot r_j$ in this rule, where $r_j$ and $r_i$ are the time averaged firing rates at the pre- and postsynaptic cells. Here, we use a simple case where $H_{ij}$ for LTP and LTD are identical, this is the simplest option and it is sufficient here, but

experimentally the Hebbian terms are more complex and are different for LTP and LTD traces (*He et al., 2015*). Our Hebbian term is also subject to rate thresholds $r_{\text{th}}$ and $r_{\text{th,FF}}$ in the recurrent and feed-forward cases, which we further discuss at the end of this section.. The parameter $T_{\text{max}}^{\text{a}}$ refers to the saturation level, $\tau^{\text{a}}$ to the characteristic decay time constant of the trace, and $\eta^{\text{a}}$ to the Hebbian activation constant. If we assume steady-state activity such that $<H_{ij}>$ is constant (a first-order approximation), we can derive effective saturation levels and effective time constants $\tilde{T}^{\text{a}}$ and $\tilde{\tau}^{\text{a}}$, which vary as a function of the Hebbian term:

$$\tilde{T}^{\text{a}} = \frac{T_{\text{max}}^{\text{a}} \langle H_{ij} \rangle}{T_{\text{max}}^{\text{a}} + \langle H_{ij} \rangle} \tag{10}$$

$$\tilde{\tau}^{\text{a}} = \frac{T_{\text{max}}^{\text{a}} \tau^{\text{a}}}{T_{\text{max}}^{\text{a}} + \langle H_{ij} \rangle} \tag{11}$$

where $\langle H_{ij} \rangle$ is the mean of Hebbian activity during trace saturation, in cases of prolonged and steady pre- and postsynaptic activation (*Huertas et al., 2016*). For increasing $\langle H_{ij} \rangle / T_{\text{max}}^{\text{a}}$, $\tilde{T}^{\text{a}}$ asymptotically approaches $T_{\text{max}}^{\text{a}}$. In our model this ratio is large for appreciable $\langle H_{ij} \rangle$, so practically speaking the traces saturate very close to $T_{\text{max}}^{\text{a}}$ (see *Figure 4—figure supplement 1*). *Equations 8 and 9*, using parameters as in *Supplementary file 1*, create traces with a fast, quickly saturating rising phase in the presence of constant activity, and a long, slow falling phase in the absence of activity. The trace dynamics in the rising and falling phases can be approximated using these effective terms to have the simple exponential forms:

$$T_{\text{F}}^{\text{a}}(t) = \tilde{T}^{\text{a}} e^{(t - t_{\text{stim}})/\tau^{\text{a}}} \tag{12}$$

$$T_{\text{R}}^{\text{a}}(t) = \tilde{T}^{\text{a}} \left(1 - e^{-t/\tilde{\tau}^{\text{a}}}\right) \tag{13}$$

where $T_{\text{F}}^{\text{a}}(t)$ is the trace dynamics in the falling phase and $T_{\text{R}}^{\text{a}}(t)$ represents the trace dynamics in the rising phase.

In this rule, synaptic weights are updated upon presentation of a global neuromodulatory signal $R(t)$. In general, this signal can be either a traditional reward signal or a 'novelty' signal that accompanies the beginning or end of an external stimulus. This neuromodulatory signal then converts eligibility traces into synaptic efficacies, consuming them in the process, via a simple subtraction of the LTD trace from the LTP trace:

$$\frac{dW_{ij}}{dt} = \eta R(t) \left(T_{ij}^{\text{p}} - T_{ij}^{\text{d}}\right) \tag{14}$$

Here, $W_{ij}$ is the synaptic efficacy between pre- and postsynaptic cells $j$ and $i$, $R(t)$ is the neuromodulatory signal, and $\eta$ is the learning rate. Following presentation of the neuromodulatory signal, the eligibility traces are 'consumed' and reset to zero, and their activation is set into a short refractory period (25 ms). Synaptic efficacies reach a steady state under the following condition:

$$\int_0^{t_{\text{trial}}} R(t) \left(T_{ij}^{\text{p}} - T_{ij}^{\text{d}}\right) dt = 0 \tag{15}$$

where $t_{\text{trial}}$ is the time at the end of the trial. In our model, each time a stimulus (whether a novel or familiar) starts or ends, a global novelty neuromodulator is released, and this acts as the 'reward' in the learning rule. As a result, synaptic efficacies update every time a stimulus is changed, according to the learning rule in *Equation 14*, until they reach the fixed point of *Equation 15*. Under the simplifying assumption that the reward function $R(t)$ is a delta function $\delta(t - t_{\text{reward}})$, this fixed point becomes:

$$T_{ij}^{\text{p}}(t_{\text{reward}}) = T_{ij}^{\text{d}}(t_{\text{reward}}) \tag{16}$$

In our model there are two classes of synaptic plasticity, plasticity of recurrent connections to learn duration, and of feed-forward connections between modules to learn order. When feed-

forward projections or external input activate a group of neurons, they cause an increase in the firing rates of those neurons, this we call the rising phase. Once input ceases, the activity in these neurons starts decaying, this decaying activity is the falling phase, and its dynamics are determined by recurrent connections. Synaptic eligibility traces follow the activity profiles, they rise or saturate during the rising phase, and decay during the falling phase. Since the traces saturate, the information relating to the feed-forward activity is contained exclusively in the rising phase of the traces, and during the falling phase, such information has been lost due to saturation. On the other hand, the information relating to the recurrent activity is contained in the falling phase, precisely because the input has ceased and the feed-forward information has been lost via saturation, so all that remains is the recurrent information. As such, learning the appropriate order via feed-forward connections needs to happen during the rising phase, and learning appropriate decay times via recurrent connections needs to happen during the falling phase. The differing activation, decay, and saturation rates/levels of the LTP and LTD traces have been chosen such that a fixed point can be reached either during the rising phase or falling phase of the traces (see *Figure 4—figure supplements 1* and *2*).

We show the fixed points obtained in simulations of sequence learning in *Figure 3—figure supplement 2* and demonstrate how learning occurs in *Figure 4—figure supplements 1* and *2*. To gain intuition of when these fixed points are obtained and how they depend on the learning parameters, we make several simplifying approximations. To solve for the falling phase fixed point, we first assume that the traces are saturated when the underlying Hebbian activity is above a certain threshold, $H_{th}$. We call the time when Hebbian activity crosses below that threshold $t_{th}$. Then, combining *Equations 12 and 16* (substituting $t_{reward}$ in for $t$ and $t_{th}$ in for $t_{stim}$), we can solve for $D = t_{reward} - t_{th}$ at the fixed point:

$$D = \frac{\ln\left(\frac{\tilde{T}^d}{\tilde{T}^p}\right)}{\left(\frac{1}{\tau^p} - \frac{1}{\tau^d}\right)} \tag{17}$$

The objective of learning, then, is to move $D$ from its starting value to the value in *Equation 17*, which is determined by the parameters in the network. The parameters can be chosen such that the fixed-point value of $D$ is arbitrarily small. As shown in *Figure 4—figure supplement 1*, this recaptures the behavior of the Timer cells.

The rising phase fixed point can be determined by combining *Equations 13 and 16* to give us an implicit function of $H$:

$$\tilde{T}^p_{ij}\left(1 - e^{t_{reward}/\tilde{\tau}^p_{ij}}\right) = \tilde{T}^d_{ij}\left(1 - e^{t_{reward}/\tilde{\tau}^d_{ij}}\right) \tag{18}$$

Since $\tilde{T}^a_{ij}$ and $\tilde{\tau}^a_{ij}$ both depend on $H$ (*Equations 8 and 9*), given $t_{reward}$, $T^a_{max}$, and $\tau^a$, $H$ at the time of reward is uniquely determined by this fixed point. Practically, this means feed-forward learning increases synaptic efficacies $W_{ij}$ in order to increase postsynaptic firing $r_i$, until $H_{ij} = r_i \cdot r_j = $ a fixed value at $t = t_{reward}$, as determined by *Equation 18* (*Figure 4—figure supplement 2*). Qualitatively, successful feed-forward learning in our model uses traces with dominant LTP in the rising phase ($\eta^p > \eta^d$) and dominant LTD in the falling phase ($\tau^d > \tau^p$) ( see *Supplementary file 1*).

Through these learning dynamics, the Timer cells learn to represent the duration of their particular element in the sequence (by decreasing $D$ to its fixed-point value), while Messengers learn to feed forward to the Timer cells in the next stimulated column (by increasing $W_{ij}$ until $H_{ij}(t_{reward})$ reaches its fixed-point value). Note that for modules that do not have overlapping activation, the Hebbian term $H_{ij}$ is zero, and therefore the associated weights $W_{ij}$ do not change. However, in the presence of noise, modules can have spurious activity overlaps which cause non-zero $H_{ij}$ and therefore potentiation of weights $W_{ij}$ which are non-sequential. This can lead to network instability and a failure to encode the presented sequence. To account for this, rate thresholds $r_{th}$ and $r_{th,FF}$ are included in the Hebbian term $H_{ij}$. By setting these thresholds above the effective noise level (see *Supplementary file 1*), the Hebbian overlap $H_{ij}$ used to activate traces ignores the random, noise-driven overlaps. Crucially, as $r_{th,FF}$ dictates the sensitivity to inter-columnar overlaps, it is a critical and necessary parameter to enable the all-to-all connectivity of the model. *Figure 3—figure supplement 2c* shows these weights approaching their fixed-point values over the course of learning. Earlier work derives these equations in more detail (*Huertas et al., 2016*).

The parameters chosen for the traces are displayed in *Supplementary file 1*. For the recurrent traces, the parameters follow the restrictions derived from analysis in an earlier publication (*Huertas et al., 2016*). The corresponding analysis for the feed-forward traces, however, is not necessarily applicable in the context of sequence learning, since multiple neuromodulator signals are active during each learning epoch. As a result, the parameters for the feed-forward traces are empirically derived. Multiple different sets were found to work, but the set in *Supplementary file 1* was the one used for all simulations included in this paper. *Figure 5—figure supplement 1* shows that the mean reported times are robust to random perturbations (ranging from +/– 20%) of the learning parameters. The perturbations are added via independently drawing each parameter randomly from a uniform distribution with bounds of 80–120% of each parameter's initial value. Note that the standard deviation increases, as expected, but is of the same order of magnitude. One hundred different random sets are used, and the resulting learned times compare favorably to trials with no parameter randomization.

The above eligibility trace learning rules are referred to as TTL. As described in a previous publication (*Huertas et al., 2016*), and demonstrated throughout this work, TTL allows for both feed-forward and recurrent learning, and as such can robustly encode temporally dependent input patterns. TTL is supported by recent experiments which have found evidence for eligibility traces existing in cortex (*He et al., 2015*), striatum (*Yagishita et al., 2014*), and hippocampus (*Bittner et al., 2017*; *Brzosko et al., 2017*). Other eligibility trace rules, such as the one-trace rule demonstrated in earlier work (*Gavornik et al., 2009*; *Gavornik and Shouval, 2011*), can also replicate the results of this paper. In general, any rule which can associate distal events/rewards, thereby solving the temporal credit assignment problem, would be likely to work with this network model. TTL was chosen for its biological realism, but the novel capabilities of this model (its ability to learn and recall both the duration and order of elements in a sequence) primarily result from the network architecture, combined with a history-dependent learning rule.

## Non-Markovian network dynamics

The three-stage network used to learn and recall non-Markovian sequences comprises the main columnar network, a highly recurrent network (reservoir), and a sparse pattern net. The dynamics of the rate-based columnar network are described above. The dynamics of the units $u_i$ in the reservoir are described by the equation:

$$\tau_{net}\frac{du_i}{dt} = -u_i + \sum_j J_{ij}\psi(r_j - \theta_m) + \sum_k W_{ik}\phi(u_k) \tag{19}$$

where $u_i$ are the firing rates of the units in the reservoir and $r_j$ are the firing rates of the Messenger cells in the main network. $J_{ij}$ is a K × n binary matrix of projections from the columnar network to the reservoir, where K is the number of units in the reservoir and n is the number of columns in the network. $J_{ij}$ is structured such that the first K/n units in the reservoir receive direct input from the Messengers in the first column, the second K/n units receive direct input from the Messengers in the second column, and so on. $W_{ik}$ are the recurrent weights of the reservoir, each of which is drawn from a normal distribution $N\left(0, \frac{g}{\sqrt{K}}\right)$, where g is the 'gain' of the network (*Rajan et al., 2010*).

$\psi$ is a piecewise linear activation function:

$$\psi(x) = \begin{cases} 0 \; if \; x \le 0 \\ x \; if \; x > 0 \end{cases} \tag{20}$$

so $\theta_m$ is the firing threshold. $\phi$ is a sigmoidal activation function:

$$\phi(x) = \tanh(x) \quad (21) \tag{21}$$

The reservoir projects to a high-dimensional sparse pattern network. Each unit in the sparse pattern net receives input from the reservoir, fed through a sparse matrix $O_{mi}$, with fraction $\rho = .04$ of entries non-zero and drawn from a normal distribution $N\left(0, \frac{g}{M}\right)$, where M is the number of units in the sparse pattern net and g is the gain, as described above. The activation of each unit $\nu_m$ of the sparse network is determined by the following:

$$\nu_{\mathrm{m}} = \Theta\left(\mathrm{r_j} - \theta_{\mathrm{m}}\right)\Theta\left(\sum_i \mathrm{O_{mi}}\phi(\mathrm{u_i}) - \theta_{\mathrm{o}}\right) \tag{22}$$

where $\Theta$ is the Heaviside function. The sparse pattern network is binary in our implementation but this is not essential, as long as its non-linear and sparse. The sparse network feeds back into the main network via weights $\mathrm{Q_{jm}}$, which are learned via a simple Hebbian rule:

$$\frac{\mathrm{dQ_{jm}}}{\mathrm{dt}} = \mathrm{H_{jm}} = \mathrm{r_j} \cdot \nu_{\mathrm{m}} \tag{23}$$

with the additional restriction $0 \leq \mathrm{Q_{jm}} \leq \mathrm{Q_{max}}$. Here, $j$ indexes units in the main network and $m$ indexes units in the sparse pattern net.

## Acknowledgements

We would like to acknowledge conversations with Marshall Hussain Shuler and Nicolas Brunel.

## Additional information

### Funding

| Funder | Grant reference number | Author |
|---|---|---|
| National Institute of Biomedical Imaging and Bioengineering | 1R01EB022891-01 | Harel Z Shouval |
| Office of Naval Research | N00014-16-R-BA01 | Harel Z Shouval |

The funders had no role in study design, data collection and interpretation, or the decision to submit the work for publication.

### Author contributions

Harel Z Shouval, Conceptualization, Software, Formal analysis, Writing - review and editing; Ian Cone, Conceptualization, Software, Investigation, Writing - original draft

### Author ORCIDs

Harel Z Shouval (iD) https://orcid.org/0000-0003-2799-1337

### Decision letter and Author response

Decision letter https://doi.org/10.7554/eLife.63751.sa1
Author response https://doi.org/10.7554/eLife.63751.sa2

## Additional files

### Supplementary files

- Supplementary file 1. Table of main model parameters. For full code, see http://modeldb.yale.

- Supplementary file 2. Table of reservoir, sparse net, and rate-based model parameters.

- Transparent reporting form

### Data availability

All software used for simulations will be available on ModelDb. http://modeldb.yale.edu/266774.

The following dataset was generated:

| Author(s) | Year | Dataset title | Dataset URL | Database and Identifier |
|---|---|---|---|---|
| Cone I, Shouval HZ | 2021 | Markovian and non-Markovian | http://modeldb.yale. | ModelDb, 266774 |

learning via biophysically realistic      edu/266774
learning rules

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
