## [Decision Letter]

**Acceptance summary:**

This paper describes how spiking neurons can learn spatio-temporal sequences with flexible duration of intermediate states. Available methods only allowed learning of sequences where duration could not be varied broadly. The proposition put forward in this work solves this long-standing problem in a convincing and in a biologically plausible way.

**Decision letter after peer review:**

[Editors’ note: the authors submitted for reconsideration following the decision after peer review. What follows is the decision letter after the first round of review.]

Thank you for submitting your work entitled "Learning precise spatiotemporal sequences via biophysically realistic learning rules in a modular, spiking network" for consideration by *eLife*. Your article has been reviewed by a Senior Editor, a Reviewing Editor, and two reviewers. The reviewers have opted to remain anonymous.

Our decision has been reached after consultation between the reviewers. The consensus was that the manuscript raises an interesting problem and provides an interesting but partial solution to it. The reviewers described many analyses that needed to be completed prior to publication. The reviewers thought that once all of that work was completed, the manuscript would be so substantially different that it would qualify as a new submission. Therefore, the consensus decision at this stage is not to consider further this manuscript for publication in *eLife*.

*Reviewer #1:*

In this study authors note that many models of sequential activity in the brain only account for the order in which neurons spike and fail to take into account the duration for which each neuron responds. This is indeed an interesting problem. Authors provide a network architecture that can solve part of the problem.

They assume that a neural activity sequence is consisted of many events. Neuron groups are tuned to these events and spike in order as these events occur. Each neuron group (Core Neural Architecture or CNA) consists of Timer, Messenger (both excitatory) and inhibitory neurons subgroups. The recurrent connections within Timer neurons encode the duration. The Messenger neurons project to the next CNA and encode the order.

An important part of the circuit is the inhibitory neurons which stop firing before the timer neurons (even though they are driven by timer neurons) and thereby release the Messenger neurons from inhibition at the end of the duration of the event. Thus, messenger neurons firing maximizes and signals the end of the “duration”.

Authors then expand the network to have a “reservoir” network and show that their scheme can also generate non-markovian sequences in which a particular neuron group is activated multiple times in the sequence.

I think the authors have raised an interesting problem and provide a solution that at least partly solves the problem. However, there are a number of shortcomings of the work which reduce the significance of the work.

1) Here authors are assuming sequences of distinct events. But to facilitate learning it is crucial that the animal is rewarded after each event. I am not convinced that every component of a sequential behavior is rewarded.

2) The key to indicate the end of duration of a sequence component, is that the inhibitory neurons stop firing before the Timer neurons. Why should this be? Since inhibitory neurons are driven by time neurons and that typically inhibitory neurons have low spike threshold one would expect the inhibitory neurons to remain active as long as Timer neurons are active. This must require fine tuning of the circuit which is beyond synaptic changes. This particular component of the model makes it appear contrived. Authors have not provided biological motivation or evidence of robustness of the mechanism. Authors finally predict that "There will be a population of inhibitory cells within each module that have firing properties similar to Timers but that decay more quickly (Figure 2A)." But is this enough? There will also be other inhibitory neuron type which will not have this exact dynamics -- will that be a problem?

3) While it is possible to have non-Markovian sequences but still each Timer can show only a single duration. Authors only make passing remark on how this could be amended but the solution is very ad-hoc and very likely will not scale.

4) The main motivation of the work is to come up with a biologically plausible model of sequence (order and durations). But the addition of reservoir and a sparse readout makes it non-biological. Moreover, authors have not specified the properties of the reservoir, specifically how does it keeps the memory of sequence of different messengers. And if the long-term memory of the messenger sequence is already in the reservoir, then why the sparse-net readout stage is needed.

5) The whole work is presented as demonstrations of key results, there is no systematic analysis of robustness of the results. I disagree that it is beyond the scope of the current work - I think that is exactly what should have been done, after all this is a computational study.

Other questions:

- If the Timer neurons are recurrently connected with excitatory neurons, how come their activity dies out and what determines the decay time constant in a timer circuit. Recurrently connected population of only excitatory neurons (timer cells) has only two fixed points: one at maximum value (and then the activity will sustain for ever) and the zero activity fixed point. So it would require fine tuning to get to the right duration. This again goes back to the issue of the robustness of the results.

- In a computational study like this it is very important to show the parameter ranges in which the demonstrated phenomenon is observed. Here it is also important to show under what conditions we can get a wide range of timings. Authors have chosen equal number of timer, messenger and inhibitory neurons - how crucial is this? Could we still get the same results if we assume 80-20 ratio of excitatory/inhibitory neurons?

- What is the min. and maximum duration for each event in the sequence that can be learned and recalled? (This goes back to the issue of the parameter ranges).

- What happens when the sequence parts overlap?

- Does the learning automatically stop or it has to be manually stopped.

- Since the three-stage model is more general, I guess that should be the one implemented in the brain. So I think authors should make predictions at that level? Where are reservoir and sparse net located and how would be the activity in these two modules.

*Reviewer #2:*

In the manuscript "Learning precise spatiotemporal sequences via biophysically realistic learning rules in a modular, spiking network," the authors propose a plastic spiking network model that learns selective temporal sequences with variable timing.

The authors propose a novel and tractable plasticity model with separate eligibility traces for LTD and LTP. The normative fixed point approach the authors employ to solve the temporal learning problem is elegant. Further, the writing is clear, the question well-motivated, and relevant literature cited correctly. However, the manuscript lacks a closer comparison to existing experimental data, and several open questions remain related to the network’s temporal dynamics and the robustness of the proposed learning mechanism.

1) From where do the long time scales in the network model come?

A crucial property of the timer cells is their slowly decaying firing rate which is the basis of the adjustable decays, presumably linked to the learning of the recurrent connections. It was not entirely clear what causes these slowly decaying rates in the model. The question arises because slow working memory like dynamics are non-trivial to get, especially, in spiking neural networks with physiological time constants. A body of literature tackled this issue with, for instance, attractor models (Amit and Brunel, 1997; Yakovlev et al., 1998), non-normal network dynamics (Goldman, 2009; Hennequin et al., 2012), or negative derivative feedback (Lim and Goldman, 2013). However, the present model does not seem to contain any intrinsically slow time constants. Perhaps the unusual combination of slow excitation ~80ms vs. fast inhibition (~10ms) results in some form of negative derivative feedback? This choice is opposite to the usual fast AMPA vs. slower GABA dynamics. While several potential mechanisms could underlie such slow ramping activity, it is essential to clarify this point and, perhaps, illustrate that the proposed learning scheme is robust to other slow timescale mechanisms.

2) How plausible is the delay activity state in the network model?

Although the authors used spiking neural network models for part of the study, the spiking network activity is neither shown nor characterized. How does the spiking activity look like in the present model? The firing rates seems rather high, which raises the question of how compatible the network activity is with cortical networks. For instance, the firing rates during the onset phase of the timer cell do look relatively high (>80Hz). Whereas, in many sensory areas spiking activity is relatively low and asynchronous irregular, possibly linked to a balanced state.

How does the activity in the present model compare to such balanced models? Would balance and the associated activity pose a problem? The manuscript would benefit if the mechanism proves robust to such presumably more realistic activity regimes.

3) Comparison to experimental data

One misses a more detailed treatment of how the proposed learning algorithm can be further verified with experimental data. Although inspired by experiments (Gavornik and Bear, 2014), a closer and perhaps more quantitative comparison to experimentally observable network activity in sensory cortices is desirable. Such verification may be challenging to accomplish solely based on LFP recordings. Maybe there are specific salient dynamical signatures that the present model predicts? One of the strong assumptions of the model is that there are two functionally different excitatory populations (timers and messengers) with a stereotypical interplay in time. How easily can such populations be recovered from (simulated) data?

[Editors’ note: further revisions were suggested prior to acceptance, as described below.]

Thank you for submitting your article "Learning precise spatiotemporal sequences via biophysically realistic learning rules in a modular, spiking network" for consideration by *eLife*. Your article has been reviewed by Ronald Calabrese as the Senior Editor, a Reviewing Editor, and three reviewers. The reviewers have opted to remain anonymous.

The reviewers have discussed the reviews with one another and the Reviewing Editor has drafted this decision to help you prepare a revised submission.

As the editors have judged that your manuscript is of interest, but as described below that additional simulations are required before it is published, we would like to draw your attention to changes in our revision policy that we have made in response to COVID-19 (https://elifesciences.org/articles/57162). First, because many researchers have temporarily lost access to the labs, we will give authors as much time as they need to submit revised manuscripts. We are also offering, if you choose, to post the manuscript to bioRxiv (if it is not already there) along with this decision letter and a formal designation that the manuscript is "in revision at *eLife*". Please let us know if you would like to pursue this option. (If your work is more suitable for medRxiv, you will need to post the preprint yourself, as the mechanisms for us to do so are still in development.)

This manuscript addresses the problem of learning sequences of events with variable durations. This is an important problem. The revised manuscript was reviewed by two new reviewers and one previous reviewer. The consensus was that the manuscript will be suitable for publication after the following two points are addressed:

1) Time constants of inhibitory neurons:

The choice of synaptic time constants (80ms for exc. and 10ms for inhibition) is very odd. There is no justification provided for these values.

The solution that was proposed by the reviewers is that to solve this issue by redifining the time scale by, say division by 5. So that what now is 80 milliseconds becomes 16 milliseconds. This has to be done consistently throughout the paper, but new simulations are not absolutely necessary. The point can be addressed by careful re-definition of time scales throughout the manuscript, if this is what the authors prefer.

2) The emergence of CNA and fine tuning of the model:

Another crucial parameter is the connections from timermessenger and inhibitory neurons messenger neurons. These have to be tuned such that messengers only fire when inhibition has gone down. These synapses do not appear to have been learned using the TTL in this model. This point can be addressed by testing robustness with respect to changes in the synaptic weights: what happens if connection weights are change by +/- 29 percent. New simulations are necessary here.

*Reviewer 1:*

One of the main points raised in the previous review was that of the fine tuning of the parameters. In this revision authors have not done much to address that concern (see below for my comments on their reply). So for now, I maintain that the model is contrived and rests on very strong assumptions for which there is little experimental evidence.

1) The issue of rewards:

Authors write: We have assumed that on every transition between external stimuli, a “novelty” signal causes a global release of a neuromodulator, which acts as a reinforcement signal (see Materials and methods).

This is a big assumption. Clearly, we don’t just make sequences of novel events. Familiar events are also added in sequences with novel events.

2) Time constants of inhibitory neurons:

The choice of synaptic time constants (80ms for exc. and 10ms for inh) is very odd as also noted by the second reviewer - but this is also crucial to the model. But there is no justification provided for these values.

3) The emergence of CNA and fine tuning of the model:

Another crucial parameter is the connections from timermessenger and inh neurons messenger neurons. These have to be tuned such that messenger only fire when inhibition has gone down - I do not think these synapses have been learned using the TTL in this model.

Authors argue that these synaptic weights can be learned as they have shown in Huerta et al. But CNA architecture is different from the model studied in Huerta et al. Three types of cells in Huerta et al., are all excitatory and there is a global inhibitory population. But for the CNA here we need two excitatory populations and one inhibitory. Moreover, the third inhibitory population cannot serve the function of global inhibition that may give rise to the timer/messenger cells (e.g. following Huerta et al.,). Furthermore, in the Huerta paper number of messenger cells are way too low but here authors have assumed that there are equal number of timer and messenger cells. Therefore, authors’ claim that they have provided a putative mechanism for the emergence of the CNA structure is not correct.

Since authors argue that "there are a number of degenerate sets of parameters" they should show some of those. In addition, since the choice of synaptic time constants is so odd, they should show that the model works when exc. synaptic time constant is smaller than inh time constant - or how close these two time constants can be. When a model has so many parameters it is important to question the robustness of the model.

Authors have shown the robustness of the learning algorithm (Figure 5—figure supplement 1) but there are other parameters in the model that are not associated with plastic synapses. How crucial are they? At the outset it is obvious that the synaptic time constant are crucial and for some reason exc. syn time constant has to be longer than inhibitory time constant. Similarly, the connectivity from timermessenger and inh-neuronsmessenger needs to be tuned very carefully.

4) The issue of other inhibitory neurons:

My question was not about the detectability. My concern is that besides the inh neurons in the CNA there will be other inhibitory neurons in the network e.g. those needed for the emergence of time/messenger populations. Would their activity cause problems?

5) Biological realism:

I agree with the authors that they have a local learning rule and the rule is biologically plausible. My concern was about using reservoir network (which is fine) and still calling the model (Figure 5) biologically realistic. I also do not understand the use of the term “reservoir”. To me it seems like an attractor network. As the authors know, in reservoir computing we need to train the readout of the “reservoir” to get a specific pattern in response to an input.*Reviewer 4:*

The paper addresses the problem of learning sequences composed of events of variable duration. This is an important problem and solution using messenger and timer cells is interesting and novel. I am less convinced of the utility of relying on reservoir computing to solve non-Markovian sequences. I wonder whether this can be done using connections between different CAN.

It is not clear whether the primary visual cortex (or a primary sensory area) is the best brain area to draw parallels with the proposed framework. Perhaps hippocampus would be more appropriate. In any case, layer 4 and 5 of primary sensory areas have very different properties and assigning them the same properties detracts from biological plausibility. There is some sequence learning in the visual cortex, but this is not the main effect in primary sensory processing.

---

## [Author Response]

[Editors’ note: the authors resubmitted a revised version of the paper for consideration. What follows is the authors’ response to the first round of review.]

Reviewer #1:[…]I think the authors have raised an interesting problem and provide a solution that at least partly solves the problem. However, there are a number of shortcomings of the work which reduce the significance of the work.1) Here authors are assuming sequences of distinct events. But to facilitate learning it is crucial that the animal is rewarded after each event. I am not convinced that every component of a sequential behavior is rewarded.

We do not claim that the animal is rewarded after every event, simply that there is some neuromodulator released during any change in the input (not “after each event”, what is an event?). This is how it was stated in the Results: “We have assumed that on every transition between external stimuli, a “novelty” signal causes a global release of a neuromodulator, which acts as a reinforcement signal (see Methods). The assumption of a temporally precise but spatially widespread neuromodulatory signal might seem at odds with common notions of temporally broad neuromodulator release, but they are indeed consistent with recent recordings in several neruomodulatory systems^40,41^.” Still, this is an assumption, which can be tested, and might be wrong. Our view is that we need to state our assumptions clearly, as we have done, we also think that this is a reasonable, though not a proven, assumption.

2) The key to indicate the end of duration of a sequence component, is that the inhibitory neurons stop firing before the Timer neurons. Why should this be? Since inhibitory neurons are driven by time neurons and that typically inhibitory neurons have low spike threshold one would expect the inhibitory neurons to remain active as long as Timer neurons are active.

This is addressed directly in the Results: “However, inhibitory cells in the module decay slightly more quickly than their Timer counterparts, thanks to shorter time constants for synaptic activation (80ms for excitatory, 10ms for inhibitory), and small Timer to Inhibitory weights (there are a number of degenerate sets of parameters which can facilitate quickly decaying Inhibitory cells^37^).” In modeling studies, it is indeed easy to generate this form of activity in the inhibitory population, and we have previously shown that such activity profiles can arise in a subpopulation of cells when the weights are chosen from a broad random distribution (Huertas et al., 2015). However, it is true that this is a putative mechanism which has not yet been validated experimentally.

This must require fine tuning of the circuit which is beyond synaptic changes. This particular component of the model makes it appear contrived. Authors have not provided biological motivation or evidence of robustness of the mechanism.

In our previous cited study (Huertas et al., 2015) we have shown that such sub-populations can arise in a network with efficacies chosen randomly from a broad distribution. It therefore does not depend on fine tuning. Further, such subpopulations have been observed experimentally in interval timing experiments, though there is not sufficient data in sequence learning experiments to determine if they arise there as well. We have provided a putative mechanism for how these different sub-populations may arise. The mechanism does not require fine tuning as suggested. However, it is true that it has not yet been validated experimentally.

Authors finally predict that "There will be a population of inhibitory cells within each module that have firing properties similar to Timers but that decay more quickly (Figure 2A)." But is this enough? There will also be other inhibitory neuron type which will not have this exact dynamics - will that be a problem?

In trying to interpret the question we find several options as to what they mean, we will therefore answer all options we can think of:

a) The ability to experimentally detect this sub-population. Of course, when any experimentalist looks for particular types of neurons (take place cells, or really any functionally specific cell type as an example), there will be plenty of other neurons the experimenter records which do not have the desired dynamics. This problem of detection is a universal problem in neuroscience, and is not unique to our prediction of this particular type of inhibitory neuron existing in visual cortex.

b) Will the exitance of other inhibitory cell types eliminate the ability of the model to perform. In Huertas et al., (2015) we showed that a wide distribution of inhibitory cell types results in a heterogeneity of excitatory cell types, some of these will be of the “Messenger” cell type. So, this heterogeneity of inhibitory cell types does not eliminate the ability to generate “Messenger” cell types. The harder problem in the context of the sequence learning model, is that only these “Messenger” type cells should have plasticity in their synaptic connections to the “Timers” of the subsequent column. For this we assume actually that there is some structure in the micro-column, so that the “Messenger” cells don’t arise in simply due to the distribution, but to some structure in the microcircuit column, or in other words, they are predestined to become “Messenger”. In that case it will be easier to mark them, and assume they have plastic synapses to Timers.

3) While it is possible to have non-Markovian sequences but still each Timer can show only a single duration. Authors only make passing remark on how this could be amended but the solution is very ad-hoc and very likely will not scale.

The ability to learn several target times within a single recurrent population is indeed currently a general problem with local learning rules. We have changed our language referring to this in the new version to: “In order for repeated elements to have different durations during recall, an appropriate learning rule must be capable of creating multiple attractors within that element’s Timer population, with each attractor triggering a different duration of activity.” Since this issue is not yet resolved for local learning rules, this challenge must still be overcome, and our lab is currently working on this.

4) The main motivation of the work is to come up with a biologically plausible model of sequence (order and durations). But the addition of reservoir and a sparse readout makes it non-biological.

Yes, our ultimate aim is to generate a biologically plausible model for sequence learning. More than that we want to identify the correct model. However, in this paper we do not make the claim that the reservoir structure itself is biologically realistic, nor is that the “main motivation of the work”. Our main focus it to identify architectural constraints that allow for local, biophysically plausible learning rules. From the Results: “To learn non-Markovian sequences, we modify the network structure while maintaining local learning rules.” and the Discussion: “We combine these two methods, using highly recurrent networks in the context of a larger architecture, and this combination allows us to maintain local and biophysically realistic learning rules.” From the very beginning of the Discussion: “In this work, we demonstrate the ability of a modular, spiking network to use biophysically realistic learning rules to learn and recall sequences, correctly reporting the duration and order of the individual elements.”

The rate-based implementation used here for the reservoir machine is indeed not biophysical. Additionally, the reviewer might still wonder if there is any biophysical realism to reservoir machines. This is indeed debatable and is a current focus of research in many labs. There have been various publications that have tried to implement RNN type architectures with spiking neurons (Abbott et al., 2016, Nicola and Clopath 2016, etc.) Whether this attempt has been successful is beyond the scope of this paper, and this might take years to resolve. However, some network with extended memory dependence is necessary for the formulation outlined here to work for non-Markovian sequences. We have chosen a simple sparse implementation of such a network to demonstrate its utility in solving this problem, not in order to show that this specific implementation is used in the brain. We have tried to clarify this in the current version of the paper.

Moreover, authors have not specified the properties of the reservoir, specifically how does it keeps the memory of sequence of different messengers.

We are not quite sure what the reviewer means by “properties of the reservoir”. However, the key property of the reservoir used here is that its current state depends on a long-term history of its activity. This is the property of reservoir machines explored extensively in the literature. In the paper we state: “For the network to learn and recall non-Markovian sequences, it must somehow keep track of its history, and use this history to inform transitions during sequence learning and recall. To this end, we include two additional “stages” to the network (Figure 5). The first is a fixed (non-learning) recurrent network, sometimes called a “reservoir” (as in reservoir computing) or a “liquid” (as in a liquid state machine)^49,50^, which receives inputs from the Messenger cells in the main columnar network. Owing to these inputs and due to its strong recurrent connectivity, the current state of the reservoir network is highly dependent on the history of network activity. Therefore, it acts as a long-term memory of the state of the columnar network.” Moreover, in the Materials and methods section all implementation details of the reservoir machine are given.

And if the long-term memory of the messenger sequence is already in the reservoir, then why the sparse-net readout stage is needed.

This is directly addressed in the Results: “The second additional stage is a high dimensional, sparse, non-linear network which receives input from the reservoir, serving to project the reservoir states into a space where they are highly separated and non-overlapping. The result is that a given “pattern” in this sparse network at time t uniquely identifies the history of the main network up to and including time t. Since these patterns are highly non-overlapping (due to the sparsity and non-linearity), a particular pattern at time t can use simple, local, and biophysically realistic Hebbian learning to connect to Timer cells firing at time t + ∆t in the main network (Direct Messenger to Timer feed forward learning is removed in this non-Markovian example).”

5) The whole work is presented as demonstrations of key results, there is no systematic analysis of robustness of the results. I disagree that it is beyond the scope of the current work -- I think that is exactly what should have been done, after all this is a computational study.

We have already addressed many aspects of the robustness in the original paper, even though we did not specifically call these results robustness. This might have caused the reviewer to disregard these results. Robustness to stochasticity is addressed directly in Figure 3—figure supplement 2. Robustness to input deviations is addressed directly in Figure 8—figure supplement 1. Many other aspects of the robustness of the learning rule, and of the CNA have been previously addressed in previous papers that are cited here (He et al. 2015, Huertas et al., 2015, 2016). However, we have further extended our robustness analysis and these results are currently shown in Figure 5—figure supplement 1 and cited in the main text. We have also changed our wording in several places to make to clearer to the readers that this model is robust to many parameter variations. We must note however, that the model has 35 parameters, we did not modify all of these in our computational study as this is not feasible

Other questions:- If the Timer neurons are recurrently connected with excitatory neurons, how come their activity dies out and what determines the decay time constant in a timer circuit. Recurrently connected population of only excitatory neurons (timer cells) has only two fixed points: one at maximum value (and then the activity will sustain for ever) and the zero activity fixed point. So it would require fine tuning to get to the right duration. This again goes back to the issue of the robustness of the results.

The activity of the Timer neurons is clearly transient. We are not at an activity fixed point, nor do we ever claim to be. Our weights reach fixed points, but these fixed points are not in the bi-stable regime of the network. An activity fixed point is a completely different thing. Our network is given a transient stimulus and produces a transient response. The timer cell network was previously comprehensively analyzed in Gavornik and Shouval, 2011, using mean field theory methods; these previous results are cited in this paper.

- In a computational study like this it is very important to show the parameter ranges in which the demonstrated phenomenon is observed. Here it is also important to show under what conditions we can get a wide range of timings. Authors have chosen equal number of timer, messenger and inhibitory neurons -- how crucial is this? Could we still get the same results if we assume 80-20 ratio of excitatory/inhibitory neurons?

We hope that our modified language, previous cited work, and additional work regarding robustness provide an answer to some of these questions as well. A different ratio of E/I cells would not fundamentally alter these results, neither would a different ratio of T and M cells.

- What is the min. and maximum duration for each event in the sequence that can be learned and recalled? (This goes back to the issue of the parameter ranges).

This is addressed directly in the Discussion of the original paper: “In combining modular, heterogenous structure with a learning rule based on eligibility traces, the model can accurately learn and recall sequences of up to at least 8 elements, with each element anywhere from ~300ms to ~1800ms in duration.” From the Results: “The network is capable of learning sequences of temporal intervals where the individual elements can be anywhere from ~300ms to ~1800ms (see Supplementary Figure 2) in duration, which agrees with the observed ranges used for training V1 circuits^1,6^.” With different sets of parameters, these ranges will likely differ, though we have good reason to believe that ranges beyond ~2000ms cannot be implemented in a stochastic spiking network, without additional mechanisms.

- What happens when the sequence parts overlap?

We are not sure exactly by what they mean by overlap, however we address overlaps directly in subsection “Learning and recalling non-markovian sequences”. In particular Figure 7 is titled: “Recall of Two Overlapping Sequences”.

- Does the learning automatically stop or it has to be manually stopped.

The phrase “fixed point” appears many times in the manuscript in relation to our learning rule. Here are a few examples: “We have used this rule because it can solve the temporal credit assignment problem, allowing the network to associate events distal in time, and because it reaches fixed points in both recurrent and feedforward learning tasks^35^” “Properly encoded durations and orders are the result of the fixed points in the learning rule, as described in the Materials and methods section and in previous publications^35,46^.” “Recurrent learning ends in a fixed point which sets the time D between the end of firing in one column and the start of firing in the next.” “Feed forward learning results in a fixed point which determines the connection strength between Messenger and Timer cells in subsequent columns.” Moreover, Supplementary Figure 2C shows the convergence to these fixed points, and in the Materials and methods section Equations 15-17 define these fixed points, and show analytical results regarding their convergence in a simple case, which has been previously analyzed and is cited here.

- Since the three-stage model is more general, I guess that should be the one implemented in the brain.

Yes, indeed the three-stage model, as the reviewer notes, is more general and can learn and reproduce more types of sequences. Indeed, behaviorally these are problems we can solve, so they must be embedded somewhere in the brain.

So I think authors should make predictions at that level? Where are reservoir and sparse net located and how would be the activity in these two modules.

For the modular network, there is some physiological data suggesting that this network resides already in primary sensory cortex. For the three-stage network, we have almost no physiological information, and therefore cannot meaningfully make specific predictions. However, some brain structures have circuit elements and connectivity that suggest they could be used for these purposes. In the original paper we state: “The reservoir and sparse network components of our three-stage model could be thought to arise from a projection from other cortical or subcortical areas. Functionally similar networks (ones that take complex, multimodal, and dynamic context and repackage it into sparse, separated patterns) have been observed in the dentate gyrus^54,55^ and the cerebellum^56,57^. However, these model components could also be thought of as part of the same cortical network, partially segregated in function but not necessarily by location.” The modular network is also a component of the three-stage network. For that network we make very clear predictions about the identity of the modular network, and its properties. The specific predictions we make about this modular network go significantly beyond the typically generic predictions made in most previous models (see Figure 8).

Reviewer #2:In the manuscript "Learning precise spatiotemporal sequences via biophysically realistic learning rules in a modular, spiking network," the authors propose a plastic spiking network model that learns selective temporal sequences with variable timing.The authors propose a novel and tractable plasticity model with separate eligibility traces for LTD and LTP. The normative fixed point approach the authors employ to solve the temporal learning problem is elegant. Further, the writing is clear, the question well-motivated, and relevant literature cited correctly. However, the manuscript lacks a closer comparison to existing experimental data, and several open questions remain related to the network’s temporal dynamics and the robustness of the proposed learning mechanism.1) From where do the long time scales in the network model come?

The long time-scales arise from the strong recurrent weights in the network, and is explicitly stated as such in the paper. From the Results:” The “Timer” cells learn strong recurrent connections with other Timer cells within the module. This strong recurrent connectivity results in long lasting transient activity, which is used to represent the duration of a given stimuli. Previous studies have analyzed in detail the relationship between recurrent connectivity and duration of resulting transient activity following a stimulus^35,36^.” We have previously extensively analyzed the dynamics of such recurrent networks, and this analysis is cited in the paper. We have tried to make this even more clear in this rewritten version of the paper.

A crucial property of the timer cells is their slowly decaying firing rate which is the basis of the adjustable decays, presumably linked to the learning of the recurrent connections. It was not entirely clear what causes these slowly decaying rates in the model.

As the reviewer implies, and as is stated in the model, the slowly decaying activity arises from recurrent connections. The ability of recurrent connections to represent the correct times arises from the learning rule. We have analyzed these aspects of the model previously in our publications (Gavornik and Shouval, 2011, Huertas et al., 2016), and try to make this even more clear in the current version of this paper.

The question arises because slow working memory like dynamics are non-trivial to get, especially, in spiking neural networks with physiological time constants. A body of literature tackled this issue with, for instance, attractor models (Amit and Brunel, 1997; Yakovlev et al., 1998), non-normal network dynamics (Goldman, 2009; Hennequin et al., 2012), or negative derivative feedback (Lim and Goldman, 2013). However, the present model does not seem to contain any intrinsically slow time constants. Perhaps the unusual combination of slow excitation ~80ms vs. fast inhibition (~10ms) results in some form of negative derivative feedback? This choice is opposite to the usual fast AMPA vs. slower GABA dynamics. While several potential mechanisms could underlie such slow ramping activity, it is essential to clarify this point and, perhaps, illustrate that the proposed learning scheme is robust to other slow timescale mechanisms.

The slow dynamics here are a property of a recurrent excitatory network. We have previously shown and mathematically analyzed such a network (Gavornik, Shouval, 2011). In order to carry out this analysis we indeed use the MFT methods developed by Brunel and Amit. Formally, this is the same type of network as a working memory network, only operating slightly below the bifurcation at which the network becomes bi-stable. The slow dynamics are a reflection of the “ghost” of this bifurcation. Synaptic plasticity acts to bring the “source” and “sink” terms to the appropriate distance such that the network decays with the appropriate time. Given the time constants used here, the stochastic spiking network can represent decays of close to 2000ms. For decays within this range we do not need additional mechanisms such as negative derivative feedback. The slow decay of the network can be achieved with normal weight matrixes, and this does not require non-normal dynamics. Additional mechanisms might extend the time ranges that the network can reproduce.

2) How plausible is the delay activity state in the network model?Although the authors used spiking neural network models for part of the study, the spiking network activity is neither shown nor characterized. How does the spiking activity look like in the present model?

The reviewer failed to notice Figure 3—figure supplement 4, which shows a spike raster of the entire network for a trial before learning, and a trial after learning. It is referenced in the Results: “Empirically, temporal accuracy of recall depends on many non-trivial factors (i.e. length of individual elements, length of entire sequence, placement of short elements near long elements, etc.), owing to the many non-trivial effects of stochasticity of the spiking network (spike rasters are shown in Figure 3—figure supplement 4).”

The firing rates seems rather high, which raises the question of how compatible the network activity is with cortical networks. For instance, the firing rates during the onset phase of the timer cell do look relatively high (>80Hz). Whereas, in many sensory areas spiking activity is relatively low and asynchronous irregular, possibly linked to a balanced state.

The cell types in our model are based off of findings of interval timing-based cells in visual cortex, such as from Liu et al., (2015). Figure 2 compares our model to these experimentally observed cell types, which also have very high firing rates (~40 Hz) for the Timers.

How does the activity in the present model compare to such balanced models? Would balance and the associated activity pose a problem? The manuscript would benefit if the mechanism proves robust to such presumably more realistic activity regimes.

The model was not set up to be in the balanced regime, however its spike statistics are quire realistic. This can be seen in the additional subplot showing the ISI distributions of Timer and Messenger cells (Figure 3—figure supplement 4). It’s actually quite intriguing and surprising that despite not being in the balanced regime, the model results in reasonable spike statistics. Partially this is due to injected noise into neurons in order to generate spontaneous activity. We primarily included this additional variability in order to make sure that our learning mechanisms are robust enough to handle such noise. We are currently analyzing the source of these surprisingly realistic spike statistics, as part of a different project.

3) Comparison to experimental dataOne misses a more detailed treatment of how the proposed learning algorithm can be further verified with experimental data. Although inspired by experiments (Gavornik and Bear, 2014), a closer and perhaps more quantitative comparison to experimentally observable network activity in sensory cortices is desirable. Such verification may be challenging to accomplish solely based on LFP recordings. Maybe there are specific salient dynamical signatures that the present model predicts? One of the strong assumptions of the model is that there are two functionally different excitatory populations (timers and messengers) with a stereotypical interplay in time. How easily can such populations be recovered from (simulated) data?

Indeed, our model predicts that in sequence learning networks would develop “Timer” and “Messenger” type populations of cells. These cells are indeed found in single unit recording in interval-timing paradigms, but not yet in sequence learning paradigms as most of the data there is based on LFPs. In our simulated data it is easy to distinguished from the data alone these two different populations, even without knowing a priori which cell belong to which group. The statistical properties of these cells even at the single cell level, such as duration of higher firing rate, and the firing rates themselves, are clearly significantly different. Indeed, we make a strong prediction that these two population will exist, and testing experimentally this is essential for experimentally validating, rejecting, or modifying the model.

[Editors’ note: what follows is the authors” response to the second round of review.]

This manuscript addresses the problem of learning sequences of events with variable durations. This is an important problem. The revised manuscript was reviewed by two new reviewers and one previous reviewer. The consensus was that the manuscript will be suitable for publication after the following two points are addressed:1) Time constants of inhibitory neurons:The choice of synaptic time constants (80ms for exc. and 10ms for inhibition) is very odd. There is no justification provided for these values.The solution that was proposed by the reviewers is that to solve this issue by redifining the time scale by, say division by 5. So that what now is 80 milliseconds becomes 16 milliseconds. This has to be done consistently throughout the paper, but new simulations are not absolutely necessary. The point can be addressed by careful re-definition of time scales throughout the manuscript, if this is what the authors prefer.

Our excitatory time constant is notably long (80 ms), but this is not strictly required for our model. We have included a new figure (Figure 5—figure supplement 2) where we set the excitatory time constant to 20ms and demonstrate successful learning of a sequence with elements 500ms long each. However, the ability of the Timers to learn long times via their recurrent connections (without prohibitively small learning rates) depends on such large time constants, which are common in working memory literature (Wang et al., 2013, Lisman et al., 1998, Gavornik and Shouval, 2011). Figure 5—figure supplement 2B shows that the Timer cells reach bistability when trying to learn 1000ms with a 20ms time constant, causing failure in learning. The relationship between reported time, recurrent weights and time constants in Timer-like cells is analyzed in detail in previous work (Gavornik and Shouval, 2011 in particular). Although there is some evidence for a slow time constant in PFC (Wang et al., 2013), this might not be so in sensory cortex. There are alternative ways to acquire a slow time constant that can facilitate learning of long interval times. Such options include derivative feedback (Lim and Goldman, 2013), and active intrinsic conductance (Fransen et al.,. 2006, Gavornik and Shouval, 2011). Such work is beyond the scope of the current paper.

2) The emergence of CNA and fine tuning of the model:Another crucial parameter is the connections from timermessenger and inhibitory neurons messenger neurons. These have to be tuned such that messengers only fire when inhibition has gone down. These synapses do not appear to have been learned using the TTL in this model. This point can be addressed by testing robustness with respect to changes in the synaptic weights: what happens if connection weights are change by +/- 29 percent. New simulations are necessary here.

These synapses were not learned using TTL in this model, but were learned using single trace learning in Huertas et al., 2015. We demonstrate that sequence learning is robust to +/- 20% changes in these synaptic weights in a new Figure (5). There is very little difference between any of the cases. The most notable is the +20% Timer to Messenger weight case, where the Messengers have a noticeably higher firing rate, but other than that, changes to the network are negligible. These weights do not have to be particularly finely tuned in order to achieve successful sequence learning.

Reviewer 1:One of the main points raised in the previous review was that of the fine tuning of the parameters. In this revision authors have not done much to address that concern (see below for my comments on their reply). So for now, I maintain that the model is contrived and rests on very strong assumptions for which there is little experimental evidence.

We hope that our responses to (1) and (2) above help assuage your concerns.

1) The issue of rewards:Authors write: We have assumed that on every transition between external stimuli, a “novelty” signal causes a global release of a neuromodulator, which acts as a reinforcement signal (see Materials and methods).This is a big assumption. Clearly, we don’t just make sequences of novel events. Familiar events are also added in sequences with novel events.

The neuromodulatory “novelty” signal acts on “every transition between external stimuli”, not just novel external stimuli (as can be seen in the non-Markovian section, where stimuli are repeated).

2) Time constants of inhibitory neurons:The choice of synaptic time constants (80ms for exc. and 10ms for inh) is very odd as also noted by the second reviewer -- but this is also crucial to the model. But there is no justification provided for these values.

See response to “Time constants of inhibitory neurons”.

3) The emergence of CNA and fine tuning of the model:Another crucial parameter is the connections from timermessenger and inh neurons messenger neurons. These have to be tuned such that messenger only fire when inhibition has gone down - I do not think these synapses have been learned using the TTL in this model.Authors argue that these synaptic weights can be learned as they have shown in Huerta et al. But CNA architecture is different from the model studied in Huerta et al. Three types of cells in Huerta et al. are all excitatory and there is a global inhibitory population. But for the CNA here we need two excitatory populations and one inhibitory. Moreover, the third inhibitory population cannot serve the function of global inhibition that may give rise to the timer/messenger cells (e.g. following Huerta et al.,). Furthermore, in the Huerta paper number of messenger cells are way too low but here authors have assumed that there are equal number of timer and messenger cells. Therefore, authors” claim that they have provided a putative mechanism for the emergence of the CNA structure is not correct.

The CNA we use here is the same as in Huertas et al., see Figure 7 inset from that paper. The “sustained decrease” (SD) cells in Figure 4D from that paper are omitted because they are not necessary for sequence learning. Huertas starts from a global excitatory and inhibitory population, but the main connections that result from trace learning (hence “core” neural architecture) results in a CNA identical to the one we show in Figure 2. It is true that we have assumed an equal number of Timers and Messengers, while Heurtas (and experimental evidence) suggest a smaller number of Messengers than Timers. Our model can still function with a smaller number of Messengers, we simply chose an equal number here for simplicity. Functionally the Messengers will operate the same as long as there are enough neurons to produce a mean response of the population with a high SNR (from some quick simulations this seems to be around 20 or so rather than 100).

Since authors argue that "there are a number of degenerate sets of parameters" they should show some of those. In addition, since the choice of synaptic time constants is so odd, they should show that the model works when exc. synaptic time constant is smaller than inh time constant - or how close these two time constants can be. When a model has so many parameters it is important to question the robustness of the model.

See responses to “Time constants of inhibitory neurons” and “The emergence of CNA and fine tuning of the model”.

Authors have shown the robustness of the learning algorithm (supp 9) but there are other parameters in the model that are not associated with plastic synapses. How crucial are they? At the outset it is obvious that the synaptic time constant are crucial and for some reason exc. syn time constant has to be longer than inhibitory time constant. Similarly, the connectivity from timermessenger and inh-neuronsmessenger needs to be tuned very carefully.

See responses to “Time constants of inhibitory neurons” and “The emergence of CNA and fine tuning of the model”. Those weights do not in fact need to be tuned especially carefully.

4) The issue of other inhibitory neurons:My question was not about the detectability. My concern is that besides the inh neurons in the CNA there will be other inhibitory neurons in the network e.g. those needed for the emergence of time/messenger populations. Would their activity cause problems?

I do not understand this question. Within each CNA there is only a single class of inhibitory interneurons. The inhibitory neurons in the CNA are the ones responsible for the emergence of the timer/messenger population.

5) Biological realism:I agree with the authors that they have a local learning rule and the rule is biologically plausible. My concern was about using reservoir network (which is fine) and still calling the model (Figure 5) biologically realistic.

We explicitly do not use the terms “biologically realistic” or “biophysically realistic” to describe the non-Markovian part of the model (nor do we use them wholesale to describe the Markovian part of the model, either). We only claim that it maintains a local learning rule. From the Results: “We have chosen this simple sparse representation of this three-stage network, not because it is a biophysically realistic implementation but in order to demonstrate the concept that such an addition is sufficient for learning and expressing non-Markovian sequences, while still using local learning rules.”

I also do not understand the use of the term “reservoir”. To me it seems like an attractor network. As the authors know, in reservoir computing we need to train the readout of the “reservoir” to get a specific pattern in response to an input.

Reservoir (or “liquid”) are general terms for the recurrent network used in reservoir computing or in a liquid state machine. We are not performing reservoir computing, but we are using a reservoir (or liquid, or RNN) in our model. From the Results: “The first is a fixed (non-learning) recurrent network, sometimes called a “reservoir” (as in reservoir computing) or a “liquid” (as in a liquid state machine)^49,50^, which receives inputs from the Messenger cells in the main columnar network.” I would not consider it an attractor network since the only fixed points are 0 and saturation (in the limit of large N and large t, but we operate far away from this regime, and therefore far away from the fixed points).

Reviewer 4:The paper addresses the problem of learning sequences composed of events of variable duration. This is an important problem and solution using messenger and timer cells is interesting and novel. I am less convinced of the utility of relying on reservoir computing to solve non-Markovian sequences. I wonder whether this can be done using connections between different CAN.

The reviewer offers an interesting alternative approach to learning non-Markovian sequences. We indeed tried approaches similar to the ones he proposes, and have not been able to get them to work. Of course, it might still be possible with different details than the ones we have tried, so this is still an interesting direction for future work. One interesting model of hierarchical non-Markovian sequence processing is Hawkins and Ahmad, 2016.

It is not clear whether the primary visual cortex (or a primary sensory area) is the best brain area to draw parallels with the proposed framework. Perhaps hippocampus would be more appropriate. In any case, layer 4 and 5 of primary sensory areas have very different properties and assigning them the same properties detracts from biological plausibility. There is some sequence learning in the visual cortex, but this is not the main effect in primary sensory processing.

The direct experimental evidence that inspires our model (the results of Gavornik and Bear, the evidence for Timer and Messenger cells) all occurs in visual cortex, hence why we draw parallels there. Hippocampus is certainly more traditionally associated with sequences, but they are often compressed, i.e. during place cell replay. Layer 4/5 were grouped here together because the deep layers appear to be functionally similar for sequence learning in Gavornik and Bear (see Figure 8D). There is some unpublished data that suggests Layer 5 as a more likely candidate for the Timer cells, so we will revise to simply layer 5. Our notation Layer 4/5 was more an indication of our uncertainty on which it was rather than implying that Timers were occurring in both.

There is also emerging evidence of Timer/Messenger type responses in other brain areas, including PFC (J. Cohen lab, unpublished), and we are currently working on related models in these brain areas.